# DeLighT: Deep and Light-weight Transformer

Sachin Mehta[1], Marjan Ghazvininejad[2], Srinivasan Iyer[2],
Luke Zettlemoyer[1,2], and Hannaneh Hajishirzi[1,3]

[1]University of Washington    [2]Facebook AI Research    [3]Allen Institute for AI

## Abstract

We introduce a deep and light-weight transformer, `DeLighT`, that delivers similar or better performance than standard transformer-based models with significantly fewer parameters. `DeLighT` more efficiently allocates parameters both (1) within each Transformer block using the `DeLighT` transformation, a deep and light-weight transformation and (2) across blocks using block-wise scaling, that allows for shallower and narrower `DeLighT` blocks near the input and wider and deeper `DeLighT` blocks near the output. Overall, `DeLighT` networks are 2.5 to 4 times deeper than standard transformer models and yet have fewer parameters and operations. Experiments on benchmark machine translation and language modeling tasks show that `DeLighT` matches or improves the performance of baseline Transformers with 2 to 3 times fewer parameters on average.

## 1 Introduction

Attention-based transformer networks (Vaswani et al., 2017) are widely used for sequence modeling tasks, including language modeling and machine translation. To improve performance, models are often scaled to be either wider, by increasing the dimension of hidden layers, or deeper, by stacking more transformer blocks. For example, T5 (Raffel et al., 2019) uses a dimension of 65K and GPT-3 (Brown et al., 2020) uses 96 transformer blocks. However, such scaling increases the number of network parameters significantly (e.g., T5 and GPT-3 have 11 billion and 175 billion parameters, respectively), and complicates learning, i.e., these models either require very large training corpora (Raffel et al., 2019; Devlin et al., 2019; Brown et al., 2020) or careful regularization (Hinton et al., 2012; Wan et al., 2013; Merity et al., 2018a). In this paper, we introduce a new parameter-efficient attention-based architecture that can be easily scaled to be both wide and deep.

Our `Deep` and `Light`-weight Transformer architecture, `DeLighT`, extends the transformer architecture of Vaswani et al. (2017) and delivers similar or better performance with significantly fewer parameters and operations. At the heart of `DeLighT` is the `DeLighT` transformation that uses the group linear transformations (GLTs) of Mehta et al. (2018) with an expand-reduce strategy for varying the width and depth of the `DeLighT` block efficiently. Since GLTs are local by nature, the `DeLighT` transformation uses feature shuffling, which is analogous to channel shuffling in convolutional networks (Zhang et al., 2018), to share information between different groups. Such wide and deep representations facilitate replacing the multi-head attention and feed-forward layers in transformers with single headed attention and light-weight feed-forward layers, reducing total network parameters and operations. Importantly, unlike transformers, the `DeLighT` transformation decouples the depth and width from the input size, allowing us to allocate parameters more efficiently across blocks by using shallower and narrower `DeLighT` blocks near the input and deeper and wider `DeLighT` blocks near the output.

We demonstrate that `DeLighT` models achieve similar or better performance than transformer models with significantly fewer parameters and operations, on two common sequence modeling tasks, (i) machine translation and (ii) language modeling. On the low resource WMT'16 En-Ro machine translation dataset, `DeLighT` attains transformer performance using $2.8\times$ fewer parameters. On the high resource WMT'14 En-Fr dataset, `DeLighT` delivers better performance (+0.4 BLEU score) with $1.8\times$ fewer parameters than baseline transformers. Similarly, on language modeling, `DeLighT` matches the performance of Transformer-XL (Dai et al., 2019) with $1.5\times$ fewer parameters

on the WikiText-103 dataset. Our source code is open-source and is available at: https://github.com/sacmehta/delight

## 2 RELATED WORK

**Improving transformers:** Several methods have been introduced to improve the transformer architecture. The first line of research addresses the challenge of computing self attention on long input sequences (Child et al., 2019; Kitaev et al., 2020; Beltagy et al., 2020). These methods can be combined with our architecture. The second line of research focuses on explaining multi-head attention (Raganato and Tiedemann, 2018; Brunner et al., 2020). They show that increasing the number of transformer heads can lead to redundant representations (Voita et al., 2019a; Michel et al., 2019) and using fixed attention heads with predefined patterns (Raganato et al., 2020) or synthetic attention matrices (Tay et al., 2020) improves performance. The third line of research focuses on improving transformers by learning better representations (Wu et al., 2019; 2020; So et al., 2019). These works aim to improve the expressiveness of transformers using different transformations – for example, using convolutions (Wu et al., 2019; Gehring et al., 2017), gated linear units (Dauphin et al., 2017), or multi-branch feature extractors (So et al., 2019; Wu et al., 2020). Our work falls into this category. Unlike previous works, we show that it is possible to efficiently allocate parameters both at the block-level using the `DeLighT` transformation and across blocks using block-wise scaling.

**Model scaling:** Model scaling is a standard method to improve the performance of sequence models (Vaswani et al., 2017; Raffel et al., 2019; Lan et al., 2020; Devlin et al., 2019; Shoeybi et al., 2019; Tan and Le, 2019; Brown et al., 2020). Model dimensions are increased in width-wise scaling (Vaswani et al., 2017; Devlin et al., 2019) while more blocks (e.g., Transformer blocks) are stacked in depth-wise scaling (Shoeybi et al., 2019; Brown et al., 2020; Wang et al., 2019). In both cases (and their combination), parameters inside each block of the network are the same, which may lead to a sub-optimal solution. To further improve the performance of sequence models, this paper introduces *block-wise scaling* that allows for variably-sized blocks and efficient allocation of parameters in the network. Our results show that (1) shallower and narrower `DeLighT` blocks near the input and deeper and wider `DeLighT` blocks near the output deliver the best performance, and (2) models with block-wise scaling coupled with model scaling achieve better performance compared to model scaling alone. We note that convolutional neural networks (CNNs) also learn shallower and narrower representations near the input and deeper and wider representations near the output. Unlike CNNs (e.g., ResNet of He et al. 2016) that perform a fixed number of operations at each convolutional layer, the proposed block-wise scaling uses a variable number of operations in each layer and block.

**Improving sequence models:** There is also significant recent work on other related methods for improving sequence models, including (1) improving accuracy using better token-level representations – for example, using BPE (Sennrich et al., 2016), adaptive inputs (Baevski and Auli, 2019) and outputs (Grave et al., 2017a), and DeFINE (Mehta et al., 2020), and (2) improving efficiency – for example, using compression (Chen et al., 2018; Sun et al., 2020), pruning (Han et al., 2016; Voita et al., 2019b), and distillation (Hinton et al., 2015; Sanh et al., 2019). The closest to our work is the DeFINE transformation, which also learns representations using an expand-reduce strategy. The key difference between the DeFINE transformation (Figure 1c) and the `DeLighT` transformation (Figure 1d) is that the `DeLighT` transformation more efficiently allocates parameters within expansion and reduction layers. Unlike DeFINE, which uses fewer groups in group linear transformations to learn wider representations, `DeLighT` transformation uses more groups to learn wider representations with fewer parameters. The `DeLighT` transformation achieves comparable performance to the DeFINE transformation but with significantly fewer parameters.

## 3 DELIGHT: DEEP AND LIGHT-WEIGHT TRANSFORMER

A standard transformer block (Figure 1a) comprises of multi-head attention that uses a query-key-value decomposition to model relationships between sequence tokens, and a feed forward network (FFN) to learn wider representations. Multi-head attention obtains query $\mathbf{Q}$, key $\mathbf{K}$, and value $\mathbf{V}$ by applying three projections to the input, each consisting of $h$ linear layers (or heads) that map the $d_m$-dimensional input into a $d_h$-dimensional space, where $d_h = d_m/h$ is the head dimension. The FFN consists of two linear layers, where the first expands the dimensions from $d_m$ to $d_f$ and the

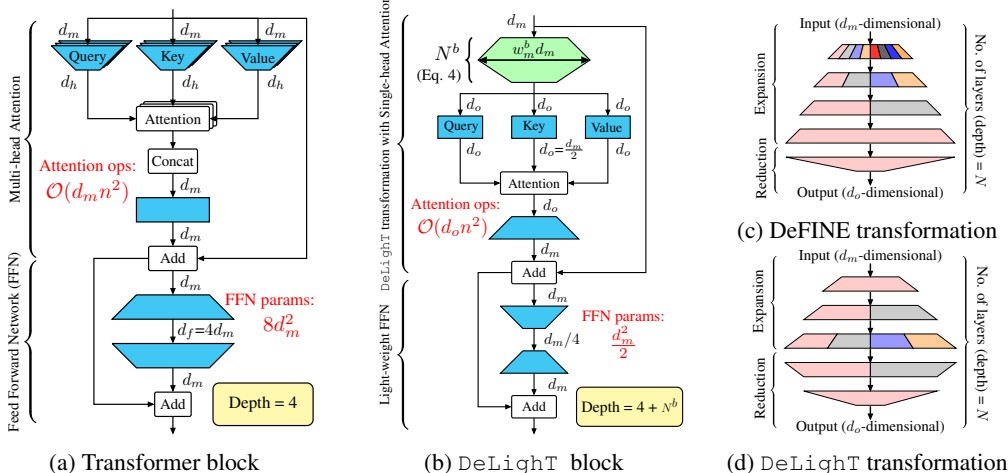

Figure 1: **(a, b)** Block-wise comparison between the standard transformer block of Vaswani et al. (2017) and the `DeLighT` block. In the `DeLighT` transformation, the number of operations in computing attention are reduced by half while the number of parameters (and operations) in the FFN are reduced by $16\times$. Transformations with learnable parameters ( Linear and `DeLighT` ) are shown in color. The shape of linear transformations indicate their operation (expansion, reduction, etc.). **(c, d)** compares the DeFINE transformation (Mehta et al., 2020) with the `DeLighT` transformation. Compared to the DeFINE transformation, the `DeLighT` transformation uses group linear transformations (GLTs) with more groups to learn wider representations with fewer parameters. Different colors are used to show groups in GLTs. For simplicity, feature shuffling is not shown in (d).

second reduces the dimensions from $d_f$ to $d_m$. The depth of a transformer block is 4, consisting of (1) three parallel branches for queries, keys, and values, (2) a fusion layer that combines the output of multiple heads, and (3) two sequential linear layers in the FFN. In general, transformer-based networks sequentially stacks transformer blocks to increase network capacity and depth.

This paper extends the transformer architecture and introduces a deep and light-weight transformer, `DeLighT`. Our model uses a deep and light-weight expand-reduce transformation, `DeLighT` transformation (Section 3.1), that enables learning wider representations efficiently. It also enables replacing multi-head attention and feed forward network (FFN) layers with single-head attention and a light-weight FFN (Section 3.2). `DeLighT` transformation decouples attention dimensions from the depth and width, allowing us to learn representations efficiently using block-wise scaling instead of uniform stacking of transformer blocks (Section 3.3).

## 3.1 DeLighT Transformation

`DeLighT` transformation maps a $d_m$ dimensional input vector into a high dimensional space (expansion) and then reduces it down to a $d_o$ dimensional output vector (reduction) using $N$ layers of the group transformations of Mehta et al. (2018), as shown in Figure 1d. During these expansion and reduction phases, `DeLighT` transformation uses group linear transformations (GLTs) because they learn local representations by deriving the output from a specific part of the input and are more efficient than linear transformations. To learn global representations, the `DeLighT` transformation shares information between different groups in the group linear transformation using feature shuffling, analogous to channel shuffling in convolutional networks (Zhang et al., 2018).

A standard approach to increase the expressivity and capacity of transformers is to increase the input dimensions, $d_m$. However, increasing $d_m$ linearly also increases the number of operations in multi-head attention ($\mathcal{O}(n^2 d_m)$, where $n$ is the sequence length) in a standard transformer block (Figure 1a). In contrast, to increase the expressivity and capacity of the `DeLighT` block, we increase the depth and width of its intermediate `DeLighT` transformations using expansion and reduction phases. This enables us to use smaller dimensions for computing attention, requiring fewer operations.

Formally, the `DeLighT` transformation is controlled by five configuration parameters: (1) number of GLT layers $N$, (2) width multiplier $w_m$, (3) input dimension $d_m$, (4) output dimension $d_o$, and

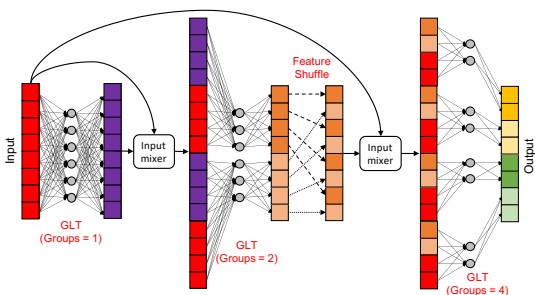

Figure 2: Example illustrating the expansion phase in the `DeLighT` transformation that uses GLTs, feature shuffling, and an input mixer connection, to learn deeper and wider representations efficiently. For illustrative purposes, we have used the same input and output dimensions.

(5) maximum groups $g_{max}$ in a GLT. In the expansion phase, the `DeLighT` transformation projects the $d_m$-dimensional input to a high-dimensional space, $d_{max} = w_m d_m$, linearly using $\lceil \frac{N}{2} \rceil$ layers. In the reduction phase, the `DeLighT` transformation projects the $d_{max}$-dimensional vector to a $d_o$-dimensional space using the remaining $N - \lceil \frac{N}{2} \rceil$ GLT layers. Mathematically, we define the output $\mathbf{Y}$ at each GLT layer $l$ as:

$$\mathbf{Y}^l = \begin{cases} \mathcal{F}\left(\mathbf{X}, \mathbf{W}^l, \mathbf{b}^l, g^l\right), & l = 1 \\ \mathcal{F}\left(\mathcal{H}\left(\mathbf{X}, \mathbf{Y}^{l-1}\right), \mathbf{W}^l, \mathbf{b}^l, g^l\right), & \text{Otherwise} \end{cases} \tag{1}$$

where $\mathbf{W}^l = \left\{\mathbf{W}_1^l, \cdots, \mathbf{W}_{g^l}^l\right\}$ and $\mathbf{b}^l = \left\{\mathbf{b}_1^l, \cdots, \mathbf{b}_{g^l}^l\right\}$ are the learnable weights and biases of group linear transformation $\mathcal{F}$ with $g^l$ groups at the $l$-th layer. Briefly, the $\mathcal{F}$ function takes the input $\mathbf{X}\left(\text{or } \mathcal{H}\left(\mathbf{X}, \mathbf{Y}^{l-1}\right)\right)$ and splits into $g^l$ non-overlapping groups such that $\mathbf{X} = \left\{\mathbf{X}_1, \cdots, \mathbf{X}_{g^l}\right\}$. The function $\mathcal{F}$ then linearly transforms each $\mathbf{X}_i$ with weights $\mathbf{W}_i^l$ and bias $\mathbf{b}_i^l$ to produce output $\mathbf{Y}_i^l = \mathbf{X}_i \mathbf{W}_i^l + \mathbf{b}_i^l$. The outputs of each group $\mathbf{Y}_i^l$ are then concatenated to produce the output $\mathbf{Y}^l$. The function $\mathcal{H}$ first shuffles the output of each group in $\mathbf{Y}^{l-1}$ and then combines it with the input $\mathbf{X}$ using the input mixer connection of Mehta et al. (2020) to avoid vanishing gradient problems. Figure 2 visualizes the expansion phase in the `DeLighT` transformation with group linear transformation, feature shuffling, and the input mixer connection.

The number of groups at the $l$-th GLT in `DeLighT` transformation are computed as:

$$g^l = \begin{cases} \min(2^{l-1}, g_{max}), & 1 \leq l \leq \lceil N/2 \rceil \\ g^{N-l}, & \text{Otherwise} \end{cases} \tag{2}$$

In our experiments, we use $g_{max} = \lceil \frac{d_m}{32} \rceil$ so that each group has at least 32 input elements.

## 3.2 DeLighT block

Figure 1b shows how we integrate `DeLighT` transformation into the transformer block to improve its efficiency. The $d_m$-dimensional inputs are first fed to the `DeLighT` transformation to produce $d_o$-dimensional outputs, where $d_o < d_m$. These $d_o$-dimensional outputs are then fed into a single head attention, followed by a light-weight FFN to model their relationships.

**`DeLighT` layer and single head attention:** Let us assume we have a sequence of $n$ input tokens, each of dimensionality $d_m$. These $n$, $d_m$-dimensional inputs are first fed to the `DeLighT` transformation to produce $n$, $d_o$-dimensional outputs, where $d_o < d_m$. These $n$, $d_o$-dimensional outputs are then projected simultaneously using three linear layers to produce $d_o$-dimensional queries $\mathbf{Q}$, keys $\mathbf{K}$, and values $\mathbf{V}$. We then model contextual relationships between these $n$ tokens using scaled dot-product attention (Eq. 3). To enable the use of residual connections (He et al., 2016), the $d_o$-dimensional outputs of this attention operation are linearly projected into a $d_m$-dimensional space.

$$\text{Attention}(\mathbf{K}, \mathbf{Q}, \mathbf{V}) = \text{softmax}\left(\frac{\mathbf{Q}\mathbf{K}^T}{\sqrt{d_o}}\right)\mathbf{V} \tag{3}$$

We hypothesize that the ability of `DeLighT` to learn wider representations allows us to replace multi-head attention with single-head attention. The computational costs for computing attention in

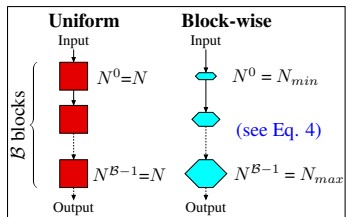 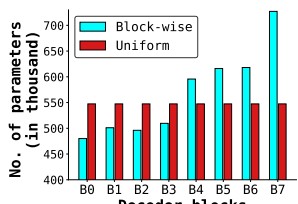 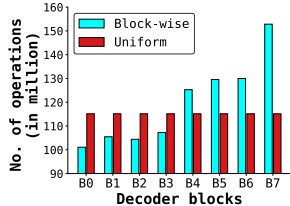

(a) Uniform vs. block-wise

(b) Distribution of parameters and operations within each block

Figure 3: **Block-wise scaling** efficiently allocates parameters and operations across blocks, leading to shallower and narrower `DeLighT` blocks near the input and deeper and wider `DeLighT` blocks near the output. In (b), `DeLighT` networks with both uniform ($N=N_{min}=N_{max}=8$) and block-wise ($N_{min}=4$, $N_{max}=8$) scaling have about 16.7 M parameters and perform 3.5 B operations (computed for a sequence length of $n = 30$), however, the `DeLighT` network with block-wise scaling delivered 2 points better perplexity.

the standard transformer and the `DeLighT` block are $\mathcal{O}(d_m n^2)$ and $\mathcal{O}(d_o n^2)$ respectively, where $d_o < d_m$. Therefore, the `DeLighT` block reduces the cost for computing attention by a factor of $d_m/d_o$. In our experiments, we used $d_o = d_m/2$, thus requiring $2\times$ fewer multiplication-addition operations as compared to the transformer architecture.

**Light-weight FFN:** Similar to FFNs in transformers, this block also consists of two linear layers. Since the `DeLighT` block has already incorporated wider representations using the `DeLighT` transformation, it allows us to invert the functionality of FFN layers in the transformer. The first layer reduces the dimensionality of the input from $d_m$ to $d_m/r$ while the second layer expands the dimensionality from $d_m/r$ to $d_m$, where $r$ is the reduction factor (see Figure 1b). Our light-weight FFN reduces the number of parameters and operations in the FFN by a factor of $r d_f/d_m$. In the standard transformer, the FFN dimensions are expanded by a factor of 4.[1] In our experiments, we used $r = 4$. Thus, the light-weight FFN reduces the number of parameters in the FFN by $16\times$.

**Block depth:** The `DeLighT` block stacks (1) a `DeLighT` transformation with $N$ GLTs, (2) three parallel linear layers for key, query, and value, (3) a projection layer, and (4) two linear layers of a light-weight FFN. Thus, the depth of `DeLighT` block is $N + 4$. Compared to the standard transformer block (depth is 4), `DeLighT` block is deeper.

## 3.3 BLOCK-WISE SCALING

Standard methods for improving the performance of sequence models include increasing the model dimensions (width scaling), stacking more blocks (depth scaling), or both. However, such scaling is not very effective on small datasets. For example, when a Transformer-Base ($d_m = 512$) network is replaced with Transformer-Large ($d_m = 1024$) on the WMT'16 En-Ro corpus, the number of parameters increases by approximately $4\times$ while the performance does not change appreciably (BLEU: 34.28 vs. 34.35). We hypothesize that this happens because scaling model width and depth allocates parameters uniformly across blocks, which may lead to learning redundant parameters. To create deep and wide networks, we extend model scaling to the block level (see Figure 3).

**Scaling the DeLighT block:** The `DeLighT` block learns deep and wide representations using the `DeLighT` transformation, whose depth and width are controlled by two configuration parameters: the number of GLT layers $N$ and the width multiplier $w_m$, respectively (Figure 3a). These configuration parameters allow us to increase the number of learnable parameters inside the `DeLighT` block independently of the input $d_m$ and output $d_o$ dimensions. Such calibration is not possible with the standard transformer block because their expressiveness and capacity are a function of the input (input dimension = number of heads $\times$ head dimension). Here, we introduce block-wise scaling that creates a network with variably-sized `DeLighT` blocks, allocating shallower and narrower `DeLighT` blocks near the input and deeper and wider `DeLighT` blocks near the output.

To do so, we introduce two network-wide configuration parameters: minimum $N_{min}$ and maximum $N_{max}$ number of GLTs in a `DeLighT` transformation. For the $b$-th `DeLighT` block, we compute

---

[1]Transformer-base uses $d_m$=512 and $d_f$=2048 while Transformer-large uses $d_m$=1024 and $d_f$=4096.

the number of GLTs $N^b$ and the width multiplier $w_m^b$ in a `DeLighT` transformation using linear scaling (Eq. 4). With this scaling, each `DeLighT` block has a different depth and width (Figure 3a).

$$N^b = N_{min} + \frac{(N_{max} - N_{min})\, b}{\mathcal{B} - 1}, \quad w_m^b = w_m + \frac{(N_{max} - N_{min})\, b}{N_{min}(\mathcal{B} - 1)}, \quad 0 \le b \le \mathcal{B} - 1 \quad (4)$$

Here, $\mathcal{B}$ denotes the number of `DeLighT` blocks in the network. We add superscript $b$ to number of GLT layers $N$ and width multiplier $w_m$ to indicate that these parameters are for the $b$-th block.

**Network depth:** The depth of transformer block is fixed, i.e., 4. Therefore, previous works (Raffel et al., 2019; Brown et al., 2020; Wang et al., 2019) have associated the depth of transformer-based networks with the number of transformer blocks. In `DeLighT`, we present a different perspective to learn deeper representations, wherein each block is variably-sized. To compute the network depth, we use the standard definition across different domains, including computer vision (e.g., ResNet of He et al. 2016) and theoretical machine learning (Telgarsky, 2016). These works measures network depth as the number of sequential learnable layers (e.g., convolution, linear, or group linear). Similarly, the depth of `DeLighT` and transformer networks with $\mathcal{B}$ blocks is $\sum_{b=0}^{\mathcal{B}-1}(N^b + 4)$ and $4\mathcal{B}$, respectively.

## 4 EXPERIMENTAL RESULTS

We evaluate the performance of `DeLighT` on two standard sequence modeling tasks: (1) machine translation (Section 4.1) and (2) language modeling (Section 4.2).

### 4.1 MACHINE TRANSLATION

**Datasets and evaluation:** We benchmark `DeLighT` models on four datasets: (1) IWSLT'14 German-English (De-En), (2) WMT'16 English-Romanian (En-Ro), (3) WMT'14 English-German (WMT'14 En-De), and (4) WMT'14 English-French (WMT'14 En-Fr). For the IWSLT'14 De-En dataset, we replicate the setup of Wu et al. (2019) and Edunov et al. (2018), which uses 160K/7K/7K sentence pairs for training, validation, and testing with a joint BPE vocabulary of about 10K tokens, respectively. For the WMT'14 English-German (En-De) dataset, we follow the setup of Vaswani et al. (2017). The dataset has 3.9M/39K/3K sentence pairs for training, validation, and testing respectively with a joint BPE vocabulary size of 44K.[2] For the WMT'14 English-French (En-Fr) dataset, we replicate the setup of Gehring et al. (2017), which uses 36M/27K/3K sentence pairs for training, validation, and testing respectively with a joint BPE vocabulary size of 44K. The performance is evaluated in terms of *BLEU* (Papineni et al., 2002) (higher is better) on the test set. We follow Wu et al. (2019) for beam search related hyper-parameters.

**Architecture:** We follow the symmetric encoder-decoder architecture of Vaswani et al. (2017) with sinusoidal positional encodings. Both the encoder and the decoder have $\mathcal{B}$ `DeLighT` blocks. Decoder blocks are identical to the encoder blocks (Figure 1b), except that they have an additional source-target single-head attention unit before the light-weight FFN. In the source-target single-head attention unit, keys and values are projections over the encoder output (full details in Appendix A). In our experiments, we use $w_m = 2$, $N_{min} = 4$, and $N_{max} = 8$ for WMT'16 En-Ro, WMT'14 En-De, and WMT'14 En-Fr; resulting in 222 layer deep `DeLighT` networks. For IWSLT'14 De-En, we used $w_m = 1$, $N_{min} = 3$, and $N_{max} = 9$ for IWSLT'14 De-En; resulting in 289 layer deep network. For simplicity, we set $\mathcal{B} = N_{max}$. We use a learnable look-up table that maps every token in the vocabulary to a 128-dimensional vector. We implement our models using Fairseq (Ott et al., 2019) and use their provided scripts for data pre-processing, training, and evaluation.

**Training:** For IWSLT'14 De-En models, we follow the setup of Wu et al. (2019) and train all our models for 50K iterations with a batch size of 4K tokens on a single NVIDIA GTX 1080 GPU. For WMT'16 En-Ro, we follow the training setup of Ghazvininejad et al. (2019) and train models for 100K iterations on 16 NVIDIA Tesla V100 GPUs with an effective batch size of 64K tokens. For WMT'14 En-De and WMT'14 En-Fr, we follow the training set-up of Wu et al. (2019) and train our models on 16 V100 GPUs for 30K and 50K iterations, respectively. We use Adam (Kingma and Ba, 2015) to minimize cross entropy loss with a label smoothing value of 0.1 during training. For a fair comparison, we trained baseline transformer models using the same training set-up.

---

[2] We use training and validation data that is compatible with the Tensor2Tensor library (Vaswani et al., 2018) in order to have fair comparisons with recent works (e.g., Evolved Transformer).

| Model | IWSLT'14 De-En | | | | WMT'16 En-Ro | | | |
|---|---|---|---|---|---|---|---|---|
| | # Params | Ratio | BLEU | Δ BLEU | # Params | Ratio | BLEU | Δ BLEU |
| Transformer (Vaswani et al., 2017) | – | – | 34.4[†] | – | 62 M | – | 34.3[‡] | – |
| Transformer (Our impl.) | 42 M | 1.0× | 34.3 | – | 62 M | 1.0× | 34.3 | – |
| DeLighT | 14 M | 0.3× | 33.8 | -0.5 | 22 M | 0.35× | 34.3 | 0.0 |
| DeLighT | 30 M | 0.7× | **35.3** | **+1.0** | 53 M | 0.85× | **34.7** | **+0.4** |

(a) Results on small corpora

| Model | WMT'14 En-De | | | | WMT'14 En-Fr | | | |
|---|---|---|---|---|---|---|---|---|
| | # Params | Ratio | BLEU | Δ BLEU | # Params | Ratio | BLEU | Δ BLEU |
| Transformer (Vaswani et al., 2017) | 62 M | – | 27.3 | – | – | 62 M | 38.1 | – |
| Transformer (Our impl.) | 67 M | 1.0× | 27.7 | – | 67 M | 1.0× | 39.2 | – |
| DeLighT | 37 M | 0.55× | 27.6 | -0.1 | 37 M | 0.55× | 39.6 | +0.4 |
| DeLighT | 54 M | 0.80× | **28.0** | **+0.3** | 54 M | 0.80× | **40.5** | **+1.3** |

(b) Results on large corpora

Table 1: **Comparison with baseline transformers on machine translation corpora**. DeLighT models require significantly fewer parameters to achieve similar performance. Here, [†] and [‡] indicate the best reported transformer baselines from Wu et al. (2019) and Ghazvininejad et al. (2019), respectively.

| | Depth | # Params | # MACs | BLEU |
|---|---|---|---|---|
| Transformer | 60 | 67 M | 11.1 B | 39.2 |
| DeLighT | 222 | 37 M | 5.6 B | 39.6 |
| DeLighT | 222 | 54 M | 8.1 B | 40.5 |

Table 2: **DeLighT networks are deep, lightweight and efficient** as compared to transformers. BLEU score is reported on the WMT'14 En-Fr dataset. To compute network depth, we count the number of sequential layers in the network (Section 3.3). We used 20 source and 20 target tokens for computing multiplication-addition operations (MACs). See Appendex C for details.

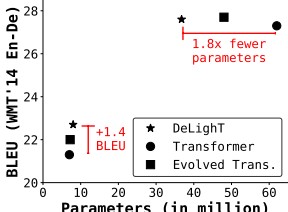

Figure 4: Comparison of DeLighT with Transformers and Evolved Transformers at two different settings, on the WMT'14 En-De corpus: (1) the number of parameters is the same and (2) the performance is the same.

### 4.1.1 RESULTS

**Comparison with baseline transformers:** Table 1 compares the performance of DeLighT with the baseline transformers of Vaswani et al. (2017) on different corpora. DeLighT delivers better performance with fewer parameters than transformers, across different corpora. Specifically, on low-resource (WMT'16 En-Ro) and high resource (WMT'14 En-De & WMT'14 En-Fr) corpora, DeLighT delivers similar or better performance with 2.8× and 1.8× fewer parameters, respectively. When the number of parameters are increased, DeLighT outperforms transformers. For example, on WMT'14 En-Fr dataset, DeLighT is 3.7× deeper than transformers and improves its BLEU score by 1.3 points yet with 13 million fewer parameters and 3 billion fewer operations (see Table 2).

Particularly interesting are the performance comparisons of DeLighT with the baseline transformers of Vaswani et al. (2017) and its neural search variant, i.e., Evolved Transformer of So et al. (2019), at two different parametric settings on WMT'14 En-De corpora in Figure 4. For small models (< 10 M parameters), DeLighT models delivers better performance and for attaining the same performance as these models, DeLighT models requires fewer parameters.

**Comparison with state-of-the-art methods:** Most state-of-the-art methods have evaluated the performance on WMT'14 En-De while some have also evaluated on IWSLT'14 De-En. Table 3 compares the performance of DeLighT with state-of-the-art methods on these two corpora. DeLighT delivers similar or better performance than existing methods. It is important to note that existing methods have improved baseline transformers with different design choices – for example, the asymmetric encoder-decoder structure (Wang et al., 2019) and neural architecture search (So et al., 2019). We believe that DeLighT, in the future, would also benefit from such design choices.

**Scaling up DeLighT models:** Figure 5 shows the performance of DeLighT models improves with increase in network parameters; suggesting their ability to learn representations across different corpora, including low-resource.

| Model | # Params | BLEU |
|---|---|---|
| Transformers (Vaswani et al., 2017) | 42 M | 34.3 |
| Variational Attention (Deng et al., 2018) | – | 33.1 |
| Dynamic convolutions (Vaswani et al., 2017) | 43 M | **35.2** |
| Lite Transformer‡ (Wu et al., 2020) | – | 33.6 |
| `DeLighT` (Ours) | **30 M** | **35.3** |

(a) IWSLT'14 De-En

| Model | # Params | BLEU |
|---|---|---|
| Transformer (Vaswani et al., 2017) | 62 M | 27.3 |
| DLCL (Wang et al., 2019) | 62 M | 27.3 |
| Evolved Transformer † (So et al., 2019) | 46 M | **27.7** |
| Lite Transformer‡ (Wu et al., 2020) | – | 26.5 |
| `DeLighT` (Ours) | **37 M** | 27.6 |

(b) WMT'14 En-De

Table 3: **Comparison with state-of-the-art methods on machine translation corpora**. `DeLighT` delivers similar or better performance than state-of-the-art models with fewer parameters. Here, † indicates that the network uses neural architecture search (NAS) and ‡ indicates that full network parameters are not reported.

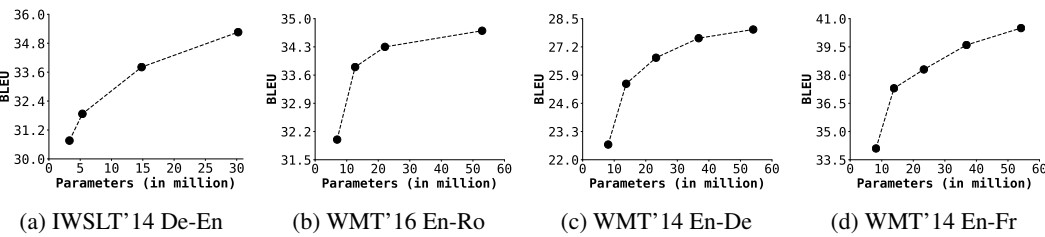

(a) IWSLT'14 De-En    (b) WMT'16 En-Ro    (c) WMT'14 En-De    (d) WMT'14 En-Fr

Figure 5: **Scaling up `DeLighT` models.** The performance of `DeLighT` improves with an increase in the number of network parameters, across different corpora, including low-resource (WMT'16 En-Ro).

## 4.2 LANGUAGE MODELING

**Datasets and evaluation:** We evaluate on the WikiText-103 dataset (Merity et al., 2017) that has 103M/217K/245K tokens for training, validation, and testing. It has a word-level vocabulary of about 260K tokens. Following recent works (Baevski and Auli, 2019; Dai et al., 2019), we report performance in terms of *perplexity* (lower is better) on the test set.

**Architecture:** We use the transformer-based decoder architecture of Baevski and Auli (2019) with $\mathcal{B}$ DeLighT blocks. We use $w_m=2$, $N_{min}=4$, and $N_{max}=12$. We scale $d_m$ using values $\{384, 512, 784, 1024\}$ for increasing network parameters. For simplicity, we set $\mathcal{B} = N_{max}$. Following standard practice, we use adaptive input (Baevski and Auli, 2019) as a look-up table and adaptive output (Grave et al., 2017a) as the classification layer with one head (head dimension is 128) and two tails (tail dimensions are 64 and 32). We also share weights between the input and the output layers.

**Training:** We follow the training setup of Baevski and Auli (2019), except that we train our models on 8 NVIDIA Tesla V100 GPUs for 100K iterations with a context length of 512 and an effective batch size of 64K tokens. We use Adam during training and use a context length of 480 during test.

**Results:** Table 4b compares the performance of `DeLighT` with previous methods on WikiText-103. Table 4a plots the variation of perplexity with number of parameters for `DeLighT` and Transformer-XL (Dai et al., 2019) – which outperforms other transformer-based implementations (e.g., Baevski and Auli 2019). Both tables show that `DeLighT` delivers better performance than state-of-the-art methods (including Transformer-XL) and it does this using a smaller context length and significantly fewer parameters, suggesting that the `DeLighT` transformation helps learn strong contextual relationships.

## 5 ANALYSIS AND DISCUSSIONS ON COMPUTATIONAL EFFICIENCY

**Training time and memory consumption:** Table 5 compares the training time and memory consumption of `DeLighT` with baseline transformers. For an apples-to-apples comparisons, we implemented the Transformer unit without NVIDIA's dedicated CUDA kernel, and trained both transformer and `DeLighT` full-precision networks for 30K iterations on 16 NVIDIA V100 GPUs. The transformer and `DeLighT` models took about 37 and 23 hours for training and consumed about 12.5 GB and 14.5 GB of GPU memory, respectively (R1 vs. R2). When we enabled the dedicated CUDA kernel provided by APEX library[3] for multi-head attention in Transformers, the training time of the

---

[3]https://github.com/NVIDIA/apex

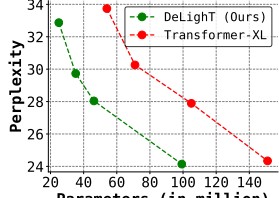

(a) DeLighT vs. Transformer-XL

(b) Comparison with existing methods

| Method | Network Depth | Context Length | # Params (in million) | Perplexity (Test) |
|---|---|---|---|---|
| LSTM (Grave et al., 2017b) | – | – | – | 48.70 |
| LSTM + Neural Cache (Grave et al., 2017b) | – | – | – | 40.80 |
| QRNN (Merity et al., 2018b) | – | – | 151 M | 33.00 |
| Transformer-XL (Dai et al., 2019) | 64 | 640 | 151 M | **24.03** |
| Transformer-XL (Our impl.)[†] | 64 | 640 | 151 M | 24.34 |
| Transformer-XL (Our impl.)[†] | 64 | 480 | 151 M | 24.91 |
| DeLighT (Ours) | 158 | **480** | **99 M** | **24.14** |

Table 4: **Results on the WikiText-103 dataset**. Compared to Transformer-XL, DeLighT delivers similar or better performance (lower perplexity) with fewer parameters. [†]For Transformer-XL, we reproduce results using the official source code. For evaluating Transformer-XL with a context length of 480, we set the mem_len hyper-parameter to 480 in the official evaluation scripts.

| Row # | Model | # Params (in million) | BLEU (WMT'14 En-Fr) | Training time | Memory (in GB) |
|---|---|---|---|---|---|
| R1 | Transformer (unoptimized) | 67 M | 39.2 | 37 hours | 12.5 GB |
| R2 | DeLighT (unoptimized) | 54 M | 40.5 | 23 hours | 14.5 GB |
| R3 | Transformer (w/ Apex optimized) | 67 M | 39.2 | 16 hours | 11.9 GB |
| R4 | DeLighT (w/ optimized grouping) | 54 M | 40.5 | 19 hours | 11.5 GB |

Table 5: Comparison with baseline transformers in terms of training speed and memory consumption. In R4, we implemented CUDA kernels for grouping and ungrouping functions only (see Appendix E). We expect DeLighT to be more efficient with a single and dedicated CUDA kernel for grouping, transformation, feature shuffling, and ungrouping. Memory consumption is measured on a single NVIDIA GP100 GPU (16 GB memory) with a maximum of 4096 tokens per batch and without any gradient accumulation.

| Model | Dropout | BLEU |
|---|---|---|
| Transformer (62 M) | 0.10 | 27.3 |
| Transformer (62 M) | 0.30 | 27.7 |
| DeLighT (37 M) | 0.05 | 27.6 |

Table 6: DeLighT requires less regularization as compared to baseline transformers (Dataset: WMT'14 En-De).

transformer model reduced from 37 to 16 hours while we did not observe any significant change in memory consumption. Motivated by this observation, we implemented dedicated CUDA kernels for grouping and ungrouping functions in GLTs (see Appendix E). With these changes, training time and GPU memory consumption of DeLighT reduced by about 4 hours and 3 GB, respectively. We emphasize that grouping, linear transformation, feature shuffling, and ungrouping, can be implemented efficiently using a single CUDA kernel. In future, we expect a dedicated CUDA kernel for these operations would further reduce the memory consumption as well as training/inference time.

**Regularization:** Table 6 shows that DeLighT delivers similar performance to baseline transformers, but with fewer parameters and less regularization. This suggests that learning representations with better transformation functions alleviates the need for dropout.

# 6 CONCLUSION

This paper introduces a deep and light-weight transformer architecture, DeLighT, that efficiently allocates parameters both within the DeLighT block and across DeLighT blocks. Compared to state-of-the-art transformer models, DeLighT models are (1) deep and light-weight and (2) deliver similar or better performance. In the future, we plan to apply DeLighT to other tasks, including language model pre-training, question answering, and language generation.

**Acknowledgements:** This research was supported by ONR N00014-18-1-2826, DARPA N66001-19-2-403, NSF (IIS-1616112, IIS1252835), and an Allen Distinguished Investigator Award. Authors would also like to thank members of the UW-NLP and the H2Lab at The University of Washington for their valuable feedback and comments.

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

## A DELIGHT ARCHITECTURES FOR LANGUAGE MODELING AND MACHINE TRANSLATION

DeLighT architectures for language modeling and machine translation are shown in Figure 6. For language modeling, we follow the architecture in Baevski and Auli (2019) while for machine translation, we follow the architecture in Vaswani et al. (2017).

**Language modeling:** Figure 6a shows the architecture for language modeling. The architecture stacks $\mathcal{B}$ DeLighT blocks, the configuration of each block is determined using block-wise scaling. Each block has three sub-layers. The first layer is a DeLighT transformation that learns representations in high-dimensional space. The second layer is a single-head attention that encodes contextual relationships. The third layer is a position-wise light-weight feed-forward network. Similar to Vaswani et al. (2017), we employ a residual connections (He et al., 2016). Similar to previous works (Baevski and Auli, 2019; Dai et al., 2019), we use tied adaptive input (Baevski and Auli, 2019) and adaptive softmax (Grave et al., 2017a) to map tokens to vectors and vectors to tokens, respectively.

**Machine translation:** Figure 6b shows the architecture for machine translation. The encoder stacks $\mathcal{B}$ DeLighT blocks, the configuration of each block is determined using block-wise scaling. Similar to language modeling, each encoder block has three sub-layers. The first layer is a DeLighT transformation that learns representations in high-dimensional space. The second layer is a single-head attention that encodes contextual relationships. The third layer is a position-wise light-weight feed-forward network. Similar to Vaswani et al. (2017), we employ a residual connections (He et al., 2016). We use learnable look-up table to map tokens

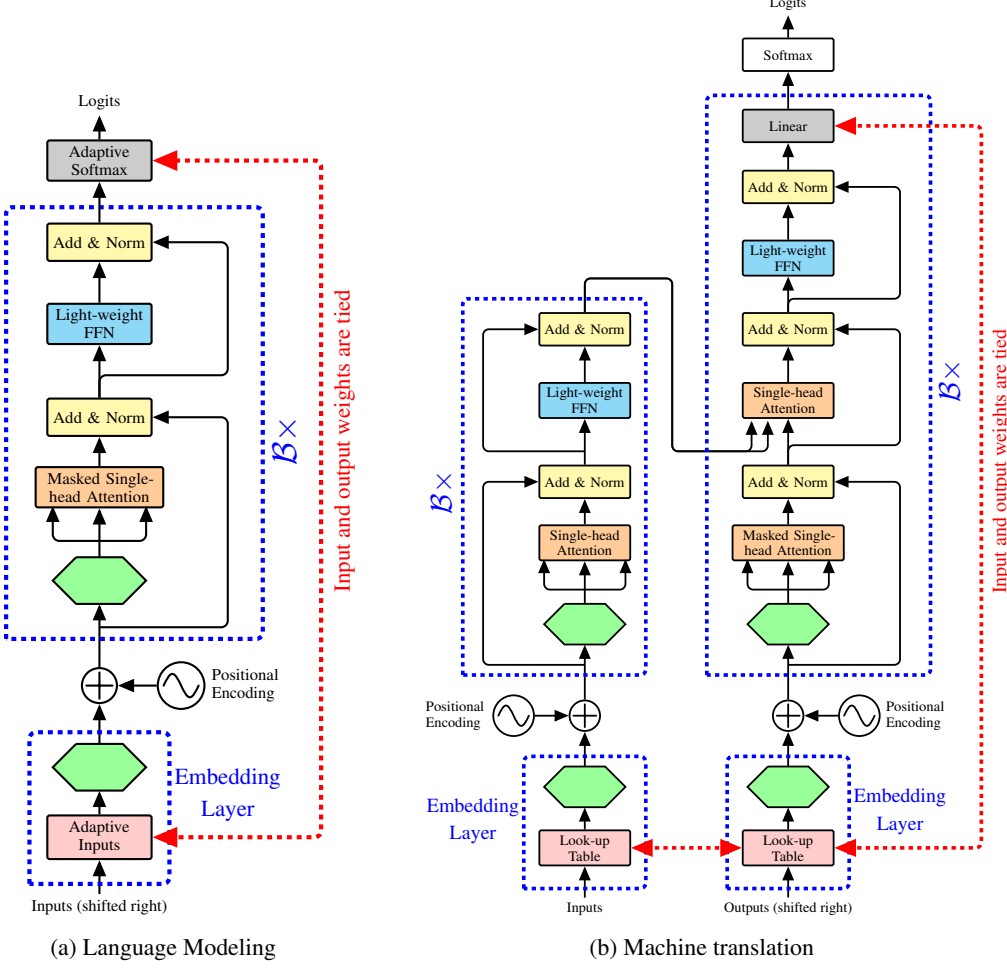

(a) Language Modeling      (b) Machine translation

Figure 6: Sequence modeling with DeLighT. Here, green color hexagon represents the DeLighT transformation.

to vectors. Similar to the encoder, the decoder also stacks $\mathcal{B}$ blocks. Decoder blocks are identical to encoder blocks, except that they have an additional source-target single-head attention unit before the light-weight FFN. Keys and values in source-target single-head attention unit are projections over the encoder output. We use standard learnable look-up table to map tokens to vectors and linear classification layer to map vectors to tokens.

## B GROUP LINEAR TRANSFORMATION WITH INPUT-MIXER CONNECTION

Group linear transformation (GLT) $\mathcal{F}$ splits a $d_m$-dimensional input $\mathbf{X}$ into $g$ non-overlapping groups such that $\mathbf{X} = \text{Concat}(\mathbf{X}_1, \cdots, \mathbf{X}_g)$, where $\mathbf{X}_i$ is a $\frac{d_m}{g}$-dimensional vector. $\mathbf{X}_i$'s are then simultaneously transformed using $g$ linear transforms $\mathbf{W}_i \in \mathbf{R}^{\frac{d_m}{g} \times \frac{d_o}{g}}$ to produce $g$ outputs $\mathbf{Y}_i = \mathbf{X}_i \mathbf{W}_i$. $\mathbf{Y}_i$'s are then concatenated to produce the final $d_o$-dimensional output $\mathbf{Y} = \text{Concat}(\mathbf{Y}_1, \cdots, \mathbf{Y}_g)$.

Figure 7a shows an example of GLT in the expansion phase of `DeLighT` transformation. For illustrative purposes, we have used the same dimensions in this example. Recall that as we go deeper in the expansion phase, the number of groups increases. In this example, the first layer has one group, the second layer has two groups and the third layer has four groups. GLTs learns group-specific representations and are local. To allow GLT to learn global representations, we use feature shuffle. An example of GLT with feature shuffle is shown in Figure 7b. Furthermore, training deep neural networks by merely stacking linear or group linear (with or without feature shuffle) is challenging because of vanishing gradient problem. Residual connections introduced by He et al. (2016) mitigates this problem and helps train deep neural networks. However, such connections cannot be employed when input and output dimensions are not the same (e.g., during the expansion and reduction phases in `DeLighT` transformation). To stabilize the training and learn deeper representations, we use input-mixer connection of Mehta et al. (2020). Figure 7c shows an example of GLT with feature shuffle and input mixer connection.

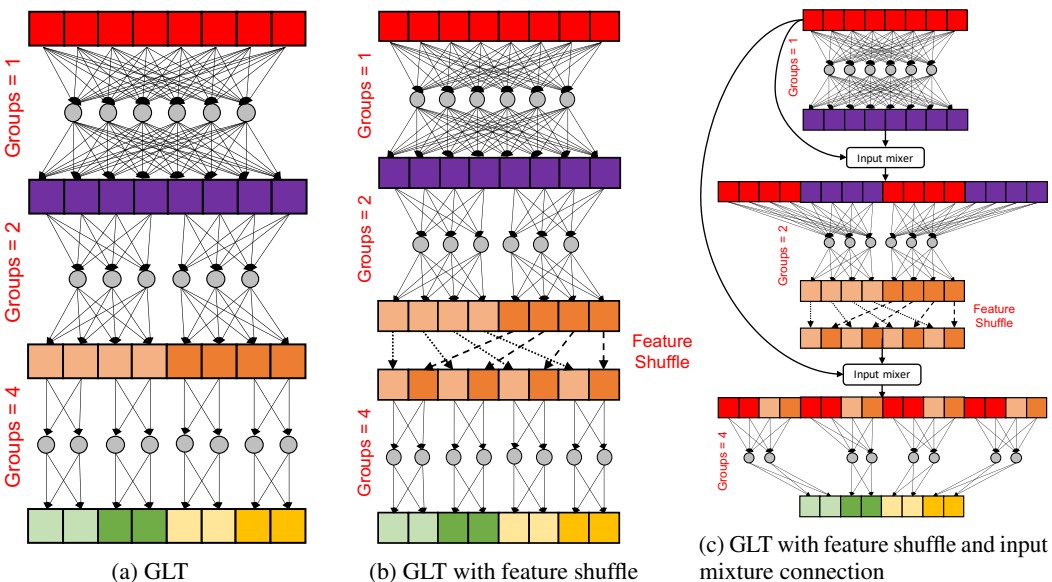

(a) GLT

(b) GLT with feature shuffle

(c) GLT with feature shuffle and input mixture connection

Figure 7: This figure visualizes different variants of group linear transformations that are used in the `DeLighT` transformation.

## C MULTIPLICATION-ADDITION OPERATIONS IN DELIGHT

The `DeLighT` block is built using linear transformations, GLTs, and scaled dot-product attention. Total number of multiplication-addition operations (MACs) in a network is an accumulation of these individual operations.

Let $n$ denotes the number of source tokens, $m$ denotes the number of target tokens, $d_m$ denotes the input dimension, $d_o$ denotes the output dimension, and $g$ denotes the number of groups in GLT. The procedure for counting MACs for each of these operations is described below.

**Group linear transformation (GLT):** GLT $\mathcal{F}$ has $g$ learnable matrices $\mathbf{W}_i \in \mathbf{R}^{\frac{d_m}{g} \times \frac{d_o}{g}}$. Therefore, GLT learns $\frac{d_m d_o}{g}$ parameters and performs $\frac{d_m d_o}{g}$ MACs to transform $d_m$-dimensional input to

$d_o$-dimensional output. Following a standard practice, e.g., ResNet of He et al. (2016), we count addition and multiplication as one operation instead of two because these operations can be fused in recent hardwares.

Importantly, when $g = 1$, the GLT is the same as linear transformation.

**Self-attention in `DeLighT`:** The scaled dot-product self-attention in `DeLighT` is defined as:

$$\text{Attention}(\mathbf{K}, \mathbf{Q}, \mathbf{V}) = \text{softmax}\left(\frac{\mathbf{Q}\mathbf{K}^T}{\sqrt{d_o}}\right)\mathbf{V} \tag{5}$$

where $\mathbf{Q} \in \mathbb{R}^{n \times d_o}$, $\mathbf{K} \in \mathbb{R}^{n \times d_o}$, $\mathbf{V} \in \mathbb{R}^{n \times d_o}$ denotes query, key, and value, respectively.

The attention operation involves two dot-products. The first dot product between $\mathbf{Q}$ and $\mathbf{K}$ while the second dot product is between the output of first dot product and $\mathbf{V}$. Both dot products require $d_o n^2$ MACs. Therefore, total number of MACs in computing scaled dot-product self-attention are $2 d_o n^2$ .

In case of a source-target attention (as in machine translation), $\mathbf{K}$'s and $\mathbf{V}$'s are from the source (encoder) and $\mathbf{Q}$'s are incrementally decoded (one token at a time). Therefore, the number of MACs required to decode $m$ target tokens given $n$ source tokens are $\sum_{k=1}^{m} 2knd_o$ .

# D ABLATIONS ON THE WIKITEXT-103 DATASET

Table 7 studies the impact of `DeLighT` block parameters on the WikiText-103 dataset, namely (1) minimum number of GLTs $N_{min}$, (2) maximum number of GLTs $N_{max}$, (3) width multiplier $w_m$, and (4) model dimension $d_m$ (see Figure 1b). Figure 8, Figure 9, and Figure 10 shows the impact of the `DeLighT` transformation, feature shuffling, and the light-weight FFN. Table 8 shows the effect of position of `DeLighT` transformation in the `DeLighT` block while Figure 12 shows the effect of scaling `DeLighT` networks. We choose the WikiText-103 dataset for ablations because it has very large vocabulary compared to other datasets (267K vs. 30-40K), allowing us to test the ability under large vocabulary sizes. The performance is reported in terms of perplexity (lower is better) on the validation set. In our ablation studies, we used the same settings for training as in Section 4.2 except that we train only for 50K iterations.

**DeLighT block:** Overall, Table 7 shows that scaling depth and width using `DeLighT` transformation and block-wise scaling improves performance. We make following observations:

a) Block-wise scaling (R4, R5) delivers better performance compared to uniform scaling (R1-R3). For instance, `DeLighT` with $N_{min} = 4$ and $N_{max} = 8$ (R4) is $1.25\times$ shallower than `DeLighT` with $N_{min} = 8$ and $N_{max} = 8$ (R2), but delivers better performance with a similar number of parameters and operations. Scaling $w_m$ improves performance (R2 vs. R3), however, the improvement is significantly lower than for the model with block-wise scaling (R3 vs. R5). This suggests that non-uniform distribution of parameters across blocks allows the network to learn better representations.

b) Different ratios between $N_{max}$ and $N_{min}$ yields different results. We observe significant performance improvements when the ratio is greater than or equal to two. For example, when we scale $\frac{N_{max}}{N_{min}}$ from 2 to 3 (R6 vs. R8), the perplexity improves by $\sim$5 points with only a moderate increase in network parameters. On the other hand, when the $\frac{N_{max}}{N_{min}}$ is close to 1 (R6 vs. R7), performance does not change appreciably. This is likely because the allocation of parameters across blocks is close to uniform (Eq. 4). This is consistent with our previous observation.

c) Learning shallower and narrower representations near the input and deeper and wider representations near the output achieves better performance. For example, when we scaled $N_{max}$ from 8 to 12 for $N_{min} = 4$ (R6, R8), `DeLighT` delivered better performance with a similar number of parameters compared to a model with $N_{min} = 6$ (R7, R9). This is likely because the ratio of $N_{max}$ and $N_{min}$ is higher when $N_{min} = 4$, which helps allocate parameters per block more effectively.

d) Deeper and wider representations near the input and shallower and narrower representations near the output hurts performance (R13 vs. R16).

e) Scaling width using $w_m$ and $d_m$ improves performance (R10-R15), however, their impact is different. For example, when we scale $w_m$ and $d_m$ by two, the rate of increase in number of parameters and operations is more rapid with $d_m$ compared to $w_m$. `DeLighT`'s ability to learn wider representations in different ways may be useful in selecting application specific models.

**Impact of `DeLighT` transformation:** We replace `DeLighT` transformation in the `DeLighT` block (Figure 1b) with (1) the DeFINE transformation and (2) a stack of linear layers. Figure 8 shows that `DeLighT` transformation delivers similar performance with significantly fewer parameters compared to the DeFINE unit

| Row # | $N_{min}$ | $N_{max}$ | $w_m$ | $d_m$ | Depth | Parameters | MACs | Perplexity |
|---|---|---|---|---|---|---|---|---|
| \multicolumn{9}{c}{**Uniform vs. block-wise scaling**} |
| R1 | 4 | 4 | 2 | 256 | 43 | 14.1 M | 2.96 B | 56.19 |
| R2 | 8 | 8 | 2 | 256 | 115 | 16.6 M | 3.49 B | 48.58 |
| R3 | 8 | 8 | 4 | 256 | 115 | 22.1 M | 4.64 B | 45.10 |
| R4 | 4 | 8 | 2 | 256 | 92 | 16.7 M | 3.51 B | 46.30 |
| R5 | 4 | 12 | 2 | 256 | 158 | 21.0 M | 4.41 B | 41.18 |
| \multicolumn{9}{c}{**Varying depth ($N_{min}$ and $N_{max}$ (Eq. 4)**} |
| R6 | 4 | 8 | 2 | 256 | 92 | 16.7 M | 3.51 B | 46.30 |
| R7 | 6 | 8 | 2 | 256 | 102 | 16.5 M | 3.46 B | 46.68 |
| R8 | 4 | 12 | 2 | 256 | 158 | 21.0 M | 4.41 B | 41.18 |
| R9 | 6 | 12 | 2 | 256 | 172 | 20.0 M | 4.20 B | 42.26 |
| \multicolumn{9}{c}{**Varying DeLighT transformation's width $w_m$ (Eq. 4)**} |
| R10 | 4 | 12 | 2 | 256 | 158 | 21.0 M | 4.41 B | 41.18 |
| R11 | 4 | 12 | 3 | 256 | 158 | 23.8 M | 4.99 B | 39.92 |
| R12 | 4 | 12 | 4 | 256 | 158 | 27.1 M | 5.69 B | 39.10 |
| \multicolumn{9}{c}{**Varying model width $d_m$**} |
| R13 | 4 | 12 | 2 | 256 | 158 | 21.0 M | 4.41 B | 41.18 |
| R14 | 4 | 12 | 2 | 384 | 158 | 29.9 M | 6.28 B | 35.14 |
| R15 | 4 | 12 | 2 | 512 | 158 | 43.8 M | 9.20 B | 30.81 |
| \multicolumn{9}{c}{**Deeper and wider near the Input**} |
| R16 | 12 | 4 | 2 | 256 | 158 | 21.0 M | 4.41 B | 43.10 |

Table 7: **Ablations on different aspects of the DeLighT block**, including uniform vs. block-wise scaling, depth scaling, and width scaling. Rows partially highlighted in color have the same configuration (repeated for illustrating results). Our experimental setup is similar to Section 4, except that we train our models for 50K iterations. Multiplication and addition operations (MACs) are computed for 20 time steps.

and linear layers. In these experiments, the settings are the same as R13-R15 (Table 7), except, $N_{max} = 8$, because models with a stack of linear layers learn too many parameters.

**Feature shuffling:** Figure 9 shows that feature shuffling improves the performance of DeLighT by 1-2 perplexity points. Here, we use the same settings as in R13-R15 (Table 7).

**Light-weight FFN:** Figure 10 shows the impact of varying the reduction factor $r$ in the light-weight FFN. We use the same settings as in R13 (Table 7). We did not observe any significant drop in performance until $r = 4$. Beyond $r = 4$, we see a drop in performance (perplexity increases by $\sim 2$ points). In such cases, the inner dimensions of the light-weight FFN are very small and hurt performance. Notably, the light-weight FFN with $r = 2^2$ delivered the same performance as $r = 2^{-2}$, but with $1.28\times$ fewer network parameters. At $r = 2^{-2}$, the light-weight FFN is the same as the FFN in Vaswani et al. (2017). This suggests that the ability of

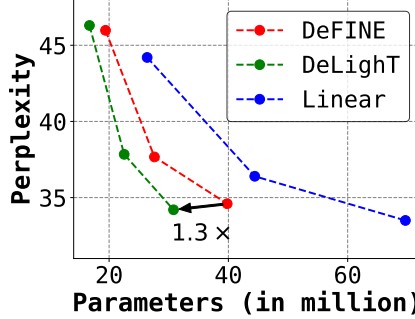

Figure 8: **Impact of different transformations.** DeLighT transformations are more parametric efficient than DeFINE and linear transformations. Lower perplexity value means better performance.

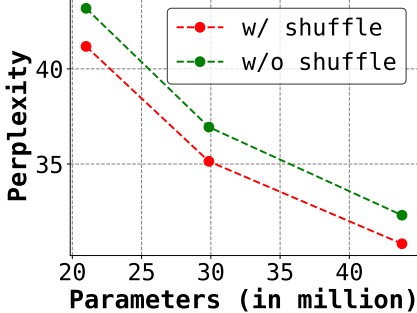

Figure 9: **Impact of feature shuffling.** Feature shuffling allows us to learn representations from global information and improves performance. Lower perplexity value means better performance.

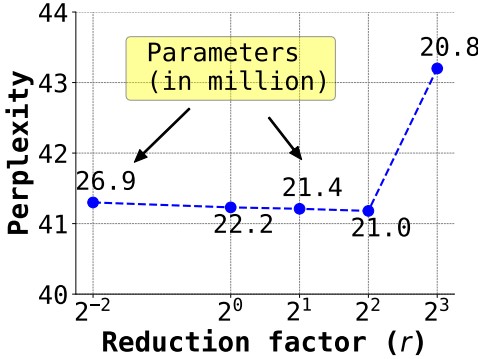

Figure 10: **Impact of reduction factor $r$ in light-weight FFN.** The ability of `DeLighT` transformation to learn representations in high-dimensional spaces efficiently allows us to reduce the computational burden on the FFN. Lower perplexity value means better performance.

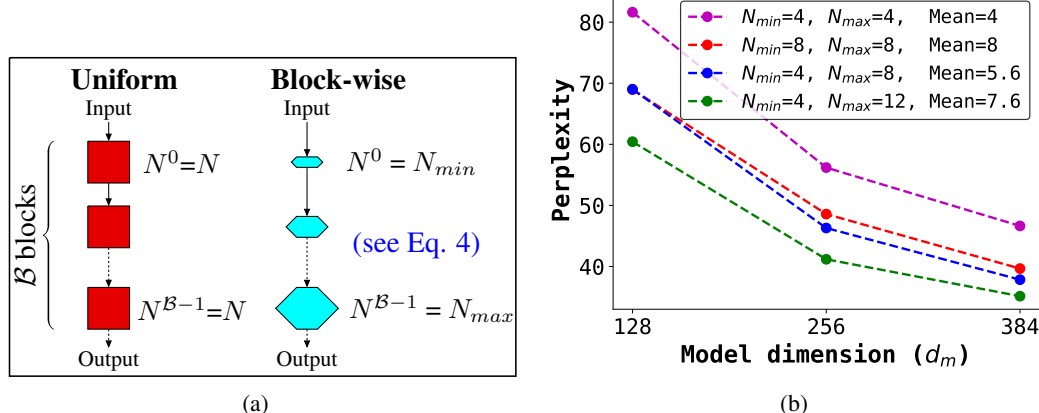

Figure 11: **Uniform vs. block-wise scaling.** (a) contrasts the uniform and block-wise scaling methods. (b) compares the results of `DeLighT` with uniform and block-wise scaling methods on the WikiText-103 dataset. `DeLighT` networks with block-wise scaling delivers better performance across different settings. Lower perplexity value means better performance.

`DeLighT` transformation to learn representations in high-dimensional spaces efficiently allows us to reduce the computational burden on the FFN.

We also tested removing the light-weight FFN and while it reduced parameters by ∼0.5-1 M, performance dropped by about 2-3 perplexity points across different parametric settings.

**Uniform vs. block-wise scaling:** Figure 11 compares the performance of `DeLighT` with uniform and block-wise scaling. For a given model dimension $d_m$, `DeLighT` models with block-wise scaling delivers better performance.

**Position of `DeLighT` transformation:**  We studied three configurations for the `DeLighT` transformation on the WikiText-103 validation set (Table 8): (1) `DeLighT` transformation followed by single-headed attention and light-weight FFN, (2) single-headed attention followed by `DeLighT` transformation, and (3) single-headed attention followed by `DeLighT` transformation and light-weight FFN. For similar number of parameters, we found that (2) and (3) drops the performance of (1) significantly across different parametric settings. This suggests that deeper and wider representations helps learn better contextual representations; allowing us to replace multi-headed attention with single-headed attention.

**Scaling up `DeLighT`:** Figure 12 shows the results of `DeLighT` models obtained after varying configuration parameters of `DeLighT` transformations ($N_{min}$={4, 6}, $N_{max}$={8, 12}, $w_m$={2, 3, 4}, and $d_m$={256, 384, 512}). We can see that scaling one configuration parameter (e.g., $d_m$) while keeping other configuration parameters constant (e.g., $N_{min}$, $N_{max}$, and $w_m$) consistently improves performance.

| Configuration | Parameters | Perplexity |
|---|---|---|
| `DeLighT` transformation + Single-head attention + Light-weight FFN | 31 M | **34.20** |
| Single-head attention + `DeLighT` transformation | 30 M | 39.02 |
| Single-head attention + `DeLighT` transformation + Light-weight FFN | 31 M | 39.43 |
| `DeLighT` transformation + Single-head attention + Light-weight FFN | 99 M | **23.16** |
| Single-head attention + `DeLighT` transformation | 96 M | 28.33 |
| Single-head attention + `DeLighT` transformation + Light-weight FFN | 99 M | 27.94 |

Table 8: **Effect of the position of `DeLighT` transformation**. Lower value of perplexity means better performance.

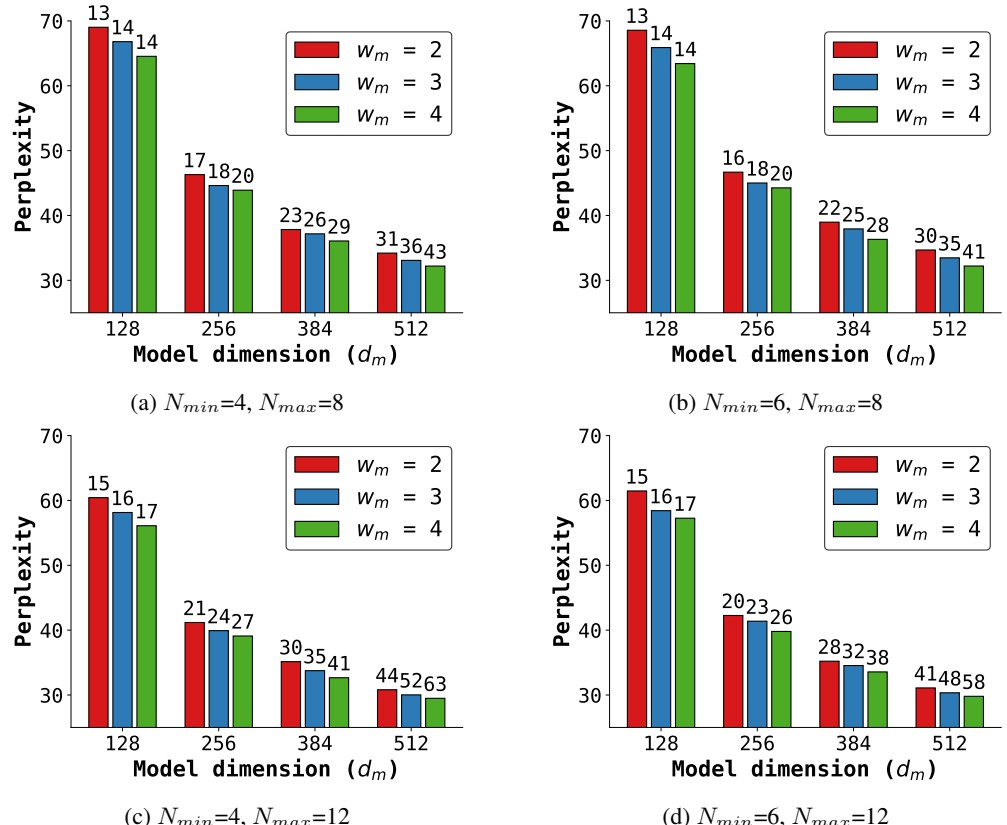

Figure 12: **Scaling up `DeLighT`**. Scaling one configuration parameter (e.g., $d_m$) while keeping other configuration parameters constant (e.g., $N_{min}$, $N_{max}$, and $w_m$) consistently improves performance. The numbers on top of each bar represents network parameters (in million). Lower value of perplexity means better performance.

This work investigates relationships between $N_{min}$, $N_{max}$, $w_m$, and $d_m$, manually. We believe that a more principled approach, such as compound scaling of Tan and Le (2019), that establishes relationships between these parameters would produce more efficient and accurate models.

# E    SOURCE CODE FOR GROUP LINEAR TRANSFORMATION

The source code for implementing group linear transformation (GLT) in PyTorch is shown in Listing 1. The source code for efficiently implementing the grouping function in GLT is shown in Listing 2. Since the ungrouping kernel is similar to grouping kernel, we have not shown it here.

The reshape and transpose operations in naive PyTorch implementation for grouping and ungrouping are replaced with a dedicated CUDA kernels, resulting in reduced memory footprint and faster training.

Listing 1: "Naive implementation of GLT in Pytorch"

```python
import torch
def glt_function(x, n_groups, weights, bias=None):
    '''
    :param x: Input tensor of size [B x N], where B is batch size and N
        is input dimension
    :param n_groups: number of groups in GLT
    :param weights: glt weights [g x N/g x M/g]
    :param bias: GLT bias (optional) of size [g x 1 x M/g]
    :return: output tensor of size [B x M]
    '''
    bsz = x.size(0)

    ## GROUPING FUNCTION: Converts [B x N] tensor to [g x B x N/g] ##
    # [B x N] --> [B x g x N/g]
    x = x.contiguous().view(bsz, n_groups, -1)
    # [B x g x N/g] --> [g x B x N/g]
    x = x.transpose(0, 1) # transpose so that group is first

    ## TRANSFORMATION FUNCTION: Transforms from N/g-dimensional space to
        M/g-dimensional space ##
    # [g x B x N/g] x [g x N/g x M/g] --> [g x B x M/g]
    x = torch.bmm(x, weights) # multiply with Weights
    # add bias
    if bias is not None:
        x = torch.add(x, bias)

    ## REGROUPING FUNCTION: Converts [g x B x M/g] tensor to [B x M] ##
    # [g x B x M/g] --> [B x g x M/g]
    x = x.transpose(0, 1) # transpose so that batch is first
    # [B x g x M/g] --> [B x M]
    x = x.contiguous().view(bsz, -1)
    return x
```

Listing 2: "Grouping kernel in CUDA"

```cpp
/* Grouping Kernel: Transforms input from [B x N] to [g x B x N/g] */
template<typename scalar_t>
__global__ void grouping_kernel_forward(const scalar_t* input,
                    const int groups, const int total_elements,
                    const int input_features, const int group_features,
                    const int batch_size, scalar_t* output){
    const int index = IMUL(blockIdx.x, blockDim.x) + threadIdx.x;
    if (index >= total_elements){
        return;
    }
    const int b_idx = index / group_features;
    const int g_f_idx = (index % group_features);
    int in_offset, out_offset;
    #pragma unroll
    for(int g=0; g < groups; g++){
        in_offset = (b_idx * input_features) + (g * group_features) +
            g_f_idx;
        out_offset = ((g * batch_size + b_idx) * group_features) + g_f_idx;
        output[out_offset] = input[in_offset];
    }
}
```

