# OpenReview forum: "DeLighT: Deep and Light-weight Transformer"
_ICLR.cc/2021/Conference — ICLR 2021 Poster_

### Official Review · AnonReviewer1 · 2020-10-26
**Less parameters is not necessarily better.**

**Rating:** 6
**Confidence:** 4

**Review:**

The paper replaces the standard MLP block with a novel building block called DeLight. DeLight is a deeper MLP of partially group-wise linear layers, which leads to parameter and potential compute savings. The authors show that their approach outperforms other transformer-like architectures with more parameters on machine translation and language modeling.

The results of the paper look encouraging at first glance but I have a couple of major concerns with this paper as it is right now:

1) The paper's motivation is that parameter rich, wide and shallow models are problematic to train due their requirement of large training data or strong regularization. However, the authors use the same training setup for both delight and standard transformers on the same datasets, and results for standard transformers are just fine, though lacking a bit behind delight. In particular, the authors do not show that using delight significantly alleviates any of the mentioned problems. They do not show that it needs less data, nor that it is "easier" to regularize.

2) The paper seems to suggest that using less parameters is better per-se, which I strongly disagree with. Especially with huge amounts of data the main driving factors for performance are compute + number of parameters. Making networks wider (instead of deep) has the advantage of massive parallel computation, which is always faster than deeper networks. That's one of the main reasons why RNNs are not used anymore in language. Furthermore, deeper models will have other issues such as storing many more activations which can result in memory bottlenecks.

3) The paper misses to discuss the most important factor which is train efficiency. In other words, how much faster can delight lead to better results than standard transformers? Note, that "larger models [can] train faster" [1], which is especially the case given large enough datasets which is definitely the case for language modeling. Unfortunately, this study optimizes for and normalizes by number of parameters. However, it should normalize by on-device training time or at least the idealized training time in flops or MACS. Right now there are some MACS comparisons but they are just for a single forward pass, not total training, which is a bit meaningless. I would like to see a proper comparison of performance to training time/flops. The best way to do this study would be to control for total training time/flops.

Finally, without showing a clear advantage in terms of training speed it is hard to judge the significance of this architectural block. I would also like to see how the differences progress when increasing the total compute equally for the vanilla transformer and delight. Will the gap between those widen or narrow the larger the models become? From my experience, the larger the models and datasets it is mostly the total compute that decides the performance, so I would actually expect the gap to become narrower.

Now I realize that the operations required by DeLight are not well supported by current accelarators (and they probably won't be in the near future), because memory shuffling is very expensive on these devices. And that's why most of the significance of this work would rely on custom kernels or even hardware. The results presented by the paper, though good, are not convincing enough for me to suggest that this customization is necessary.

[1] https://arxiv.org/abs/2002.11794

== UPDATE after rebuttal ==

I am willing to update my recommendation to weak accept after the rebuttal. Some of my main concerns remain, but added experiments, comparisons and discussions alleviate some of those. I still believe that there is too much focus on fairly useless metrics such as number of parameters which might lead to future work that will follow this trend, but the results are overall strong enough to warrant acceptance.

---

> ### Author Response · Authors · 2020-11-12
> **DeLighT requires less regularization and smaller context length**
>
> We thank you for your comment. The following tables shows the effect of regularization and context length on the performance of DeLighT and Transformers. DeLighT delivers similar performance with fewer parameters and less regularization. We observe similar phenomenon on other datasets too. Also, we observe that DeLighT requires less context during inference. We will include these details in the paper.
>
> Table: Effect of dropout on WMT'14 En-De dataset
>
> | Model | Dropout | BLEU |
> | :-----    | :-----:        | :----: |
> | Transformer (62 M) | 0.10 | 27.3 |
> | Transformer (62 M) | 0.30 | 27.7 |
> | DeLighT (37 M) | 0.05 | 27.6 |
>
> Table: Effect of context length (Task: Language modeling; Dataset: WikiText-103)
>
> | Model	| Context length	| Perplexity|
> | :-----    | :-----:        | :----: |
> | Transformer-XL (151 M params)	| 480 | 	27.32|
> | Transformer-XL (151 M params) | 640 | 	24.34 |
> | DeLighT (99 M params) | 480	| 24.14 |

---

> > ### Comment · AnonReviewer1 · 2020-11-16
> > **RE**
> >
> > The numbers for 480 context length transformer look quite high. Given the response above it seems that there are some differences in this setup and the original Transformer XL setup. Could you clarify what those are?

---

> > > ### Author Response · Authors · 2020-11-17
> > > **Re**
> > >
> > > For a direct comparison, we trained and evaluated Transformer-XL [R1] with settings similar to the one in FairSeq [R2] (DeLighT follows [R2] for training and evaluation, as noted in the paper). With these settings, we obtained a perplexity of about 27.32 for Transformer-XL with a context length of 480. However, when we evaluated official Transformer-XL model [R1], its perplexity dropped by 0.57 point when context length is changed from 640 to 480. We will clarify this in our paper.
> > >
> > > In our paper, we compared DeLighT with the best Transformer-based implementation (Transformer-XL). Note that the vanilla Transformer model with a context length of 480 gets a perplexity of 30+ (see [R2]), which can be further improved by 2-3 points with hyper-parameter tuning. In our limited experiments, we were able to improve it to 28.
> > >
> > >
> > > | Model | Set-up | Context length | Perplexity (Test) |
> > > | :----- | :-----: | :-----: | :-----: |
> > > | Transformer-XL |  Transformer-XL |  640 |   24.34 |
> > > | Transformer-XL |  Transformer-XL |  480 |   24.91 |
> > >  | Transformer-XL | Fairseq |  480 |  27.32 |
> > >  | DeLighT |  Fairseq |  480 |   **24.14** |
> > >
> > >
> > > [R1] Transformer-XL: https://github.com/kimiyoung/transformer-xl/tree/master/pytorch
> > >
> > > [R2] Fairseq: https://github.com/pytorch/fairseq/tree/master/examples/language_model#2-train-a-language-model

---

> > > > ### Comment · AnonReviewer1 · 2020-11-17
> > > > **RE**
> > > >
> > > > Wow, that's quite a difference. Note that entire papers are written because of such gaps. I think it would be good to stay transparent wrt this.

---

> ### Author Response · Authors · 2020-11-12
> **Wider networks are better**
>
> Thanks for your concern. We agree that DeLighT has many sequential layers and in theory, sequential networks are slower than wider networks. However, wider networks requires more computational resources as compared to deeper networks because during inference, activations are not stored for the entire network. This is also evident from large success of deep convolutional networks, wherein deeper networks are preferred over wider networks to learn representations, even for edge devices (e.g., MobileNets [r1, r2]). Also, it has been shown theoretically that deeper networks learns better representations over wider networks [r3].
>
> [r1] MobileNetv2: https://arxiv.org/abs/1801.04381
>
> [r2] MobileNetv3: https://arxiv.org/abs/1905.02244
>
> [r3] Why Deep Neural Networks for Function Approximation? https://arxiv.org/abs/1610.04161

---

> > ### Comment · AnonReviewer1 · 2020-11-16
> > **RE**
> >
> > Wider networks don't require "more" resources if you just control for compute. Doubling the network depth is the same as multiplying the width by sqrt(2).
> >
> > Memory is usually less of an constraint during inference and more during training. We can increase total compute by 2 while width only scales with sqrt(2) and hence memory only scales by sqrt(2). When increasing depth, the memory increases linearly with the compute.
> >
> > Again, CNNs need to be deep because of their limited receptive field, something that's not necessary in Transformers.

---

> > > ### Author Response · Authors · 2020-11-17
> > > **Re: DeLighT's memory consumption during training**
> > >
> > > During training, our unoptimized DeLighT model (Dataset: WMT'14 En-Fr, max. tokens: 4096, Depth: 222, BLEU: 40.5; Parameters: 54 M) and optimized Transformer model (Dataset: WMT'14 En-Fr, max. tokens: 4096, Depth: 60, BLEU: 39.2; Parameters: 67 M) on a single GP100 GPU with 16 GB memory consumes a memory of about **14.2 GB and 12 GB** during the first epoch, respectively. Though DeLighT is much deeper than a Transformer, its memory consumption is reasonable. This is because DeLighT efficiently allocates resources within and across blocks. As noted in our general response, we believe that dedicated CUDA kernels will further reduce the memory footprint and also improve the training/inference speed.

---

> ### Author Response · Authors · 2020-11-12
> **Why light-weight networks?**
>
> Thanks for your concern. However, we disagree with your statement ``"With huge amounts of data the main driving factors for performance are compute + number of parameters". As an example, ALBERT shows that BERT-level performance can be achieved with fewer parameters. For example, on the SQUAD dataset, ALBERT improves the performance of BERT-Large by about 2 points, yet with fewer parameters (ALBERT: 235 M; BERT: 334 M). Similarly, there is a lot of work in other ML related sub-fields, especially computer vision where it has been shown that deeper networks with better transformation functions (depth-wise convolution in MobileNets vs. standard convolution in ResNets) yields similar or better performance compared to heavy-weight and wider networks (e.g., ResNet and Wide-ResNet), that too with fewer parameters and low latency.
>
> Similar to these related fields, we also show that better transformation functions can help achieve better performance with fewer parameters and operations. Also, some NLP corpora are low-resource (e.g., En-Ro dataset that was studied in the paper). Therefore, it is important to have networks that scales well on low-, medium-, and high-resource corpora. Also, we hope that our efforts in designing light-weight and efficient models would be impactful and valuable to the community, especially resource constrained devices.
>
> We would like to highlight that there is a significant ENGINEERING effort that has led to the scalability of Transformers and we believe that DeLighT would also benefit from such efforts.
>
> [r1] ALBERT: https://arxiv.org/abs/1909.11942
>
> [r2] MobileNetv2: https://arxiv.org/abs/1801.04381
>
> [r3] ResNet: https://arxiv.org/abs/1512.03385
>
> [r4] Wide-ResNet: https://arxiv.org/abs/1605.07146
>
> [r5] BERT: https://arxiv.org/abs/1810.04805

---

> > ### Comment · AnonReviewer1 · 2020-11-16
> > **RE**
> >
> > My point is not that fewer params are worse. My point is that parameter count doesn't matter per se. What matters most is compute (except when you scale to very large models where params don't fit into memory anymore). See [1], for instance. Depth in transformers is not better than width. Generalization is generally better with larger models. ConvNets need to be deep because of their limited receptive field per layer.
> >
> > I still don't understand what engineering went specifically into transformers. Could you refer me to those? Again they are just matmuls which is efficient on all accelerators without any engineering.

---

> > > ### Author Response · Authors · 2020-11-17
> > > **Re: Depth is important in transformers**
> > >
> > > We disagree with your statement "``depth in transformers is not better than width." Several previous works have shown that deeper transformer models are better than wider transformer models. For example, DLCL [R1], a deeper version of Transformer-base, outperforms Transformer-Large with fewer parameters.
> > >
> > > Transformers are also difficult to train at larger scale (both  deep and wide) on smaller or lower-resource corpora (as we noted in our paper; Section 3.3). The work [1] that you are referring to uses a large training set. However, many NLP applications do not have such large corpora. That is why there is significant effort from research community to develop methods (e.g., different types of dropouts) that allows deep models to scale well on low-, medium-, and large-scale corpora, and this remains an ``"open" challenge. DeLighT allocates parameters and operations inside each block efficiently, and shows how we can scale neural sequence models across different corpora, including low-resource.
> > >
> > > **Regarding memory:** During training, our unoptimized DeLighT model (Dataset: WMT'14 En-Fr, max. tokens: 4096, Depth: 222, BLEU: 40.5; Parameters: 54 M) and optimized Transformer model (Dataset: WMT'14 En-Fr, max. tokens: 4096, Depth: 60, BLEU: 39.2; Parameters: 67 M) on a single GP100 GPU with 16 GB memory consumes a memory of about **14.2 GB and 12 GB** during the first epoch, respectively. Though DeLighT is much deeper than a Transformer, its memory consumption is reasonable. This is because DeLighT efficiently allocates resources within and across blocks. As noted in our general response, we believe that dedicated CUDA kernels will further reduce the memory footprint and also improve the training/inference speed.
> > >
> > > **Regarding engineering efforts**: See our response "Dedicated cuda kernels for Transformers and engineering effort"
> > >
> > > [R1] https://arxiv.org/pdf/1906.01787.pdf

---

> > > > ### Comment · AnonReviewer1 · 2020-11-17
> > > > **RE**
> > > >
> > > > Well, it seems that "fewer" parameters is then a necessity introduced by DeLIGHT (bc of memory constraints), which I wouldn't see as an advantage. I think the default setup will be to pretrain large models on a lot of data (probably self-supervised) and then finetune on whatever downstream task. This is what's already happening. I strongly believe that most of these architectural advances will therefore not "survive", because they introduce complexity with very limited benefits, especially in the large data regime. However, this is obviously just speculation and will not affect my decision.
> > > >
> > > > My position on depth vs. width remains the same, but again this shouldn't affect my final decision if the paper's formulation about their contributions and motivation is adapted. See above comment for more details.

---

> ### Author Response · Authors · 2020-11-12
> **Computational efficiency**
>
> We thank your for your feedback. Please see our general response on computational efficiency.

---

> ### Author Response · Authors · 2020-11-12
> **Shuffling is not hardware friendly**
>
> We thank you for your concern. Research in computer vision has shown that re-ordering data elements using shuffling (e.g., channel-shuffle [r1] and pixel-shuffle [r2]) can be efficiently done, including resource constrained devices. Moreover, from engineering perspective, feature shuffling operation can be done within the same group linear transformation kernel (similar to ShuffleNetv2, which also performs the concatenation and shuffling operation using the same kernel).
>
> We agree that custom kernels are required for DeLighT. However, that is the case for Transformers too. The recent success of Transformers that we see today is primarily because of significant engineering efforts from industry. We believe that DeLighT would also benefit from such engineering effort.
>
> [r1] ShuffleNetv2: https://arxiv.org/abs/1807.11164
>
> [r2] PixelShuffle: https://arxiv.org/abs/1609.05158

---

> > ### Comment · AnonReviewer1 · 2020-11-16
> > **RE**
> >
> > Thanks for the pointers!
> >
> > I am not sure whether custom kernels are needed for efficient transformers, I never used those and they are quite fast without those. Ultimately, they are almost exclusively comprised of matmuls. So in my understanding it is not True that "significant" engineering went into the development of transformers.

---

> > > ### Author Response · Authors · 2020-11-17
> > > **Re: Dedicated cuda kernels for Transformers and engineering effort**
> > >
> > > Engineering efforts in transformers can be broadly grouped into three categories: (1) GEMM optimization, (2) multi-head attention optimization, and (3) distributed training optimization (scaling up transformers).
> > >
> > > **Optimizing matrix multiplication (GEMM):** The basic transformation function in most NLP models (LSTMs or Transformers or others) is a linear or fully connected layer. This layer uses matrix multiplication operations to transform a N-dimensional vector to M-dimensional vector. There has been a significant effort from industry, especially NVIDIA, to optimize this operation and in general, hardware specific kernels are also released to support architectural changes in hardware (e.g., matrix multiplication on tensor cores). Please see CUTLASS and cuBLAS library, for example. Though these efforts are generic, NLP models, especially Transformers, significantly benefited from them.
> > >
> > > **Optimizing multi-head attention:** There is a significant effort from industry to optimize multi-head attention unit. For example, see the APEX library from NVIDIA.
> > >
> > > **Scaling transformer models:** Again, there is a significant effort from industry in scaling the transformer models and allowing training on extremely large corpora. Examples include Megatron from NVIDIA, Fairseq from Facebook, and DeepSpeed from Microsoft. Note that training such large models (e.g., Megatron) is yet infeasible in academic environment.
> > >
> > > Such engineering efforts won't be noticeable with existing deep learning frameworks, which integrates most of these developments in the backend and allows users to access these structures using simple APIs (e.g., torch.nn.MultiheadAttention). For example, multi-head attention kernel from APEX is part of PyTorch v1.7.0 now.
> > >
> > > [R1] CUTLASS and cuBLAS Library: https://github.com/NVIDIA/cutlass and https://developer.nvidia.com/cublas
> > >
> > > [R2] APEX: https://github.com/NVIDIA/apex/tree/master/apex/contrib/multihead_attn
> > >
> > > [R3] Megatron from NVIDIA: https://github.com/NVIDIA/Megatron-LM
> > >
> > > [R4] Fairseq from Facebook: https://arxiv.org/abs/1806.00187
> > >
> > > [R5] DeepSpeed from Microsoft: https://github.com/microsoft/DeepSpeed
> > >
> > > =========================================================================================
> > >
> > > **Regarding DeLighT**: The basic building block of DeLighT is a group linear transformation and at the core of it is also a dense matrix multiplication. In these transformations, small fully-connected layers are applied to non-overlapping subset of inputs. In our experiments (for research purposes), we used a solution that reshapes and transposes the tensor and is shown below (all operations are supported by current frameworks and hardwares). However, in a dedicated CUDA kernel, each subset can be processed in parallel and would increase the work done per thread, leading to faster computation. Let us assume that we want to transform a 512-dimensional input to 1024-dimensional. In case of a standard fully connected layer, this involves multiplying a 512-dimensional vector with a large dense matrix of size 512x1024. In case of a group linear transformation with 8 groups, this means we need to use 8 ``dense matrices" of size 64x128. All of these matrices can be processed using the same CUDA kernel. This would lead to (1) increased work done per thread, which will improve the GPU occupancy, (2) reduced memory footprint because reshape and transpose operations will no longer be required, and (3) faster training and inference because of fewer kernel launches.
> > >
> > > ```python
> > >
> > > import torch
> > >
> > > def glt_function(x, n_groups, weights, bias=None):
> > >     '''
> > >     :param x: Input tensor of size [B x N], where B is batch size and N is input dimension
> > >     :param n_groups: number of groups in GLT
> > >     :param weights: glt weights [g x N/g x M/g]
> > >     :param bias: GLT bias (optional) of size [g x B x M/g]
> > >     :return: output tensor of size [B x M]
> > >     '''
> > >     bsz = x.size(0)
> > >     # [B x N] --> [B x g  x N/g]
> > >     x = x.contiguous().view(bsz, n_groups, -1)
> > >     # [B x g x N/g] --> [g x B  x N/g]
> > >     x = x.transpose(0, 1)  # transpose so that group is first
> > >     # [g x B  x N/g] x [g x N/g x M/g] --> [g x B x M/g]
> > >     x = torch.bmm(x, weights)  # multiply with Weights
> > >     # add bias
> > >     if bias is not None:
> > >         x = torch.add(x, bias)
> > >     # [g x B x M/g] --> [B x g x M/g]
> > >     x = x.transpose(0, 1)  # transpose so that batch is first
> > >     # [B x g x M/g] --> [B x M]
> > >     x = x.contiguous().view(bsz, -1)
> > >     return x
> > >
> > > ```

---

> > > > ### Comment · AnonReviewer1 · 2020-11-17
> > > > **RE**
> > > >
> > > > I guess my point was that these things are not necessary for transformers to be efficient (enough) on these accelerators, s.t. a proper comparison is possible. This has been addressed in a comment, and should end up in the paper. I just wondered why these comparisons were not done in the first place.

---

> ### Comment · AnonReviewer1 · 2020-11-17
> **Update after discussions with the authors.**
>
> I am willing to rethink my recommendation based on the discussions with the authors. I would ask them to put more information into the paper:
> -  proper runtime comparisons as done in the comments here
> -  a discussion about memory requirements of their model compared to transformers (DeLIGHT can fit less parameters and this trade-off should be made clear and discussed)
> - reformulation of the motivation: Reducing number of parameters is a side-effect but shouldn't be considered a main metric to optimize for
> - comparisons in the results section should be based on apples to apples runtime comparisons, as provided in the comments here
>
> Although I have my objections, I think the results are strong enough to warrant acceptance if the paper makes these changes to increase its transparency and comparability.

---

### Official Review · AnonReviewer4 · 2020-10-27
**A thorough empirical investigation on parameter-efficient Transformers, though missing discussions on latency**

**Rating:** 6
**Confidence:** 4

**Review:**

The original Transformer includes many design heuristics, e.g., the arrangement of layers and how layer widths are tied ($d_{ffn} = 4 \cdot d_{model}$, $d_k=d_q=d_v$, etc). While much effort has been spent on up-scaling, it is also important to question these basic design heuristics.

This work does just that and presents a more parameter-efficient Transformer variant by combining existing techniques (GLT, Mehta et al, 2018; Channel Shuffling, Zhang et al. 2018) and exploring some known trade-offs (feedforward net ratio, nheads vs dhead) under the new architecture. The resulting model, DeLighT, while more complicated to describe, performs on par with the original Transformer with far fewer trainable parameters. The ablation studies seem thorough and most design choices are more or less accounted for.

My main concern is the complete absence of any discussions on time-efficiency, or latency. The only reference I could find is in the conclusion: “we expect a dedicated CUDA kernel for DeLighT units to be much more efficient, both in terms of speed as well as memory during forward and backward passes.” This implies that the current implementation might not be efficient speed-wise. However, even after excluding the benefits from CUDA kernels, this new architecture appears to be less parallelizable as it is significantly deeper than other Transformer variants. This is crucial as neural network accelerators are becoming increasingly available, even in edge devices. Before fully recommending this paper for acceptance, I’d like to see more discussions on how the proposed changes affect latency, a latency benchmark against other relevant Transformer variants, especially with parallelism, and some concrete statements regarding latency in the abstract and the conclusion. I would still consider this work valuable even if the result on latency is not strongly positive, but the work seems incomplete without any discussions of it.

Minor issues:
1)     In table 4, Delight has a context length of 480. Why not 640 for a more direct comparison?
2)     There’s little detail on how the hyperparameters are selected for the ablation study. Are they swept to account for the change in optimal hyperparameters with architectural change?

---

> ### Author Response · Authors · 2020-11-12
> **Computational efficiency**
>
> We thank you for your comment and positive response to our paper. We acknowledge your comments about computational efficiency and discuss them in general response.

---

> ### Author Response · Authors · 2020-11-12
> **Minor issues**
>
> A. Direct comparison with Transformer-XL
>
> Delight requires smaller  context length of 480 to achieve similar performance to transformer-XL (with context length 640), but uses less parameters. Following your suggestion, for a direct comparison with Transformer-XL, we report  its performance with a context length of 480 in the following table. We will update these details in the paper.
>
> |  Model | Context length   | Perplexity |
> | :------| :-------: |:------:|
> | Transformer-XL (151 M params)   |  480 |  27.32 |
> | Transformer-XL  (151 M params)   |  640 |  24.34 |
> | DeLighT (99 M params) |  480 |  24.14 |
>
> B. Hyper-parameters for the ablation studies
>
> We thank you for noting this. We did not do grid or linear search for finding optimal hyper-parameters. We used the hyper-parameters in the Fairseq repository for training attention-based models (as noted in Section 4.1; Architecture), with an exception to learning rate. Following Vaswani et al., we also scaled learning rate linearly to train models at different model dimension. We will clarify this in the paper. We will also release our source code and scripts for training and evaluation for reproducibility.

---

> > ### Public Comment · ~Anlin_Qu1 · 2020-11-14
> > **Is there a problem with the experimental results ？**
> >
> > I've personally reproduced Transformer-XL(151M) as well. But my experiments show that if context-length is set to 480, Transformer-XL's results don't drop to this level at all! So I'm wondering if the author has filled in the wrong data. I get 23.75 ppl
> > when Transformer-XL context-length is set to 640 and 23.96 when context-length is set to 480.

---

> > > ### Author Response · Authors · 2020-11-14
> > > **Experimental results on Transformer-XL**
> > >
> > > For reproducing results at context length 640, we used the source code released by the authors.
> > >
> > > But for a “direct comparison” with Transformer-XL, we used the same setup that we used for our experiments (slightly different from Transformer-XL paper). That is why our numbers are lower for context length 480. We will clarify this in our paper.
> > >
> > > Also, are your reproduced numbers on test set?

---

### Official Review · AnonReviewer3 · 2020-10-28
**Impact on  decoding in practice**

**Rating:** 7
**Confidence:** 4

**Review:**

This paper presents a variant of Transformer where low-dimension matrix multiplications and single-head attention are used. Stacked group-linear-transformation (GLT) are applied on input of each layer to perform dimension growth and then reduction. The paper is well-written and easy to follow. Experiments demonstrate the propose architecture matches or improves the performance of baseline Transformers with fewer parameters.

Although the proposed architecture has fewer MACs, it would be interesting to know the real decoding time. Deep models could reduce model size or improve performance, but layers have to be executed consecutively and may slow down decoding in practice especially when using deep decoders. I’m also curious about the performance when keeping decoders unchanged.

---

> ### Author Response · Authors · 2020-11-12
> **Computational efficiency**
>
> Thanks for supporting our paper.
>
> Please see general response regarding computational efficiency.

---

### Official Review · AnonReviewer5 · 2020-11-06
**limited contribution, but the claimed goal is achieved; very borderline**

**Rating:** 6
**Confidence:** 4

**Review:**

The paper proposes a new transformer model called DeLighT. DeLighT differs from the original transformer in the following ways:
- a DeLighT transformation is performed before the self-attention
- the self-attention is performed in a vector space with fewer dimensions and with only 1 head
- the MLP’s hidden layer is 16 times smaller
- The DeLighT transformation, as far as I understand, is a sequence of block-sparse linear layers and permutations. The number of intermediate layers in DeLightT transformations is higher in the upper layers of the whole model.

The key claims of the paper are that DeLight performs as well as or better than Transformer, while being deep, having less parameters and involving fewer floating point operations. With regard to depth, I don’t think that depth can be the goal in and of itself, it is the means for achieving good performance. I think the paper puts too much emphasis on DeLight depth. I also find it rather controversial that the depth in the paper counts all linear layers, and not just the number of nonlinearities between the input and the output.

DeLighT does seem to have a consistent edge over the original Transformer model in terms of both performance and memory footprint. Compared to TransformerXL and EvolvedTransformer, DeLighT performs on par, but has less parameters. The set of experiments performed to assert DeLighT strengths is quite extensive.  The experimental side of paper appears technically sound.

It is important to build more efficient and compact models, but I find the paper’s analysis of models computational demands rather incomplete. If computational efficiency is the main focus of the paper (and it appears to be), it would be great to see some analysis of how much time and memory is required for model training and inference. Is the model size the dominating factor here?

Overall, the architecture ends up being quite complicated. The supplementary material contains an extensive set of ablations justifying many of the complexities, but I find that a few straight-forward remain unanswered:
- how does DeLighT compare to a Transformer with one head
- how does DeLighT compare to a Transformer with a narrow MLP
- what about both above modifications?
I think it is important for the paper to justify the use of the DeLighT layer.

With regard to writing, the paper is generally quite clear and accurately done. A few places raised my eyebrows:
- “Since the DeLighT block learns wider representations of the input using the DeLighT transformation, it enables us to replace multi-head attention with single-head attention” - I think the causation here is not obvious and needs to be proven. Or the wording should be changed to illustrate that this is just your intuition.
- “This is likely because scaling model width and depth allocates parameters uniformly across blocks, which may lead to learning redundant parameters.” - same, I think it is important to state that this is a hypothesis
- I think the “input mixer” layer and how it widens the state should be discussed in the main text. Otherwise the role of w_m is completely unclear.

The paper’s novelty appears quite limited. DeLightT layers seem similar to the DeFINE layer. The difference is not sufficiently explained neither in the main text nor in the appendix. Overall it seems that the paper takes existing components and does a lot of tweaking to yield a better model with good performance.

It is hard to make a call in the case of this paper.
Pros:
- the final moment model does deliver some improvement
- the execution is quite accurate
Cons:
- the paper feels very incremental
- the papers leaves an impression that its key goals could be achieved in a simpler way
- compute time and runtime memory footprint are not discussed.

UPD: the score was updated after the rebuttal stage.

---

> ### Author Response · Authors · 2020-11-11
> **Depth of the network**
>
> We follow a standard definition for computing network depth, as noted in Section 3.3. We did not account for non-linearities and normalization layers because they are often fused with the transformation layers.

---

> ### Author Response · Authors · 2020-11-11
> **Extensive experiments**
>
> We thank you for your positive response.

---

> ### Author Response · Authors · 2020-11-11
> **Ablations with respect to Transformers (# of head and MLP)**
>
> Vaswani el al. ablated Transformers with different number of heads and MLP configurations, including single head. They found that Transformers worked best with 8 or 16 heads. Based on their empirical observations, we did not ablate the number of heads, MLP, and their combination in Transformers. For example, the performance of Transformer (base) drops by about 1 BLEU point when single-head attention is used instead of multi-head attention (8 heads). See Table 3 in Vaswani el al.'s paper: https://arxiv.org/pdf/1706.03762.pdf

---

> ### Author Response · Authors · 2020-11-11
> **Suggestions**
>
> A. “Since the DeLighT block learns wider representations of the input using the DeLighT transformation, it enables us to replace multi-head attention with single-head attention” - I think the causation here is not obvious and needs to be proven. Or the wording should be changed to illustrate that this is just your intuition.
>
> Thanks for your suggestion. We will rephrase it
>
>
> B. “This is likely because scaling model width and depth allocates parameters uniformly across blocks, which may lead to learning redundant parameters.” - same, I think it is important to state that this is a hypothesis
>
> We thank you for your suggestions and we will clarify it in the paper. Based on our ablation experiments (Table 5 and Section D in the appendix), we hypothesize that uniform scaling is not optimal. We found that allocating different parameters in each block of the model improves performance. Importantly, we found that learning deeper and wider representations near the output are more important than shallower and narrower representations near the output.
>
>
> C. I think the “input mixer” layer and how it widens the state should be discussed in the main text. Otherwise the role of w_m is completely unclear.
>
> We thank you for your suggestion. We discussed it in Appendix B along with visual illustrations. We will bring those details in the next version in Section 3.

---

> ### Author Response · Authors · 2020-11-12
> **Computational efficiency**
>
> Please see general response regarding computational efficiency.

---

> ### Author Response · Authors · 2020-11-12
> **Contributions**
>
> Most previous efforts on improving the performance of transformers are using model scaling, wherein deeper and wider transformer models are built by stacking more transformer units and using large model dimensions. However, there is little or no effort on understanding the design heuristics of transformers (as noted by AnonReviewer4). Moreover, several previous studies have shown that multi-head attention learns redundant representations (as discussed in Section 2).
>
> This work uses the existing transformation functions (group linear transformations) and show how to learn deeper and wider representations with dot-product attention efficiently. DeLighT more efficiently allocates parameters both (1) within
> each Transformer block using the DeLighT transformation, a deep and lightweight transformation and (2) across blocks using block-wise scaling, that allows for shallower and narrower DeLighT blocks near the input and wider and deeper DeLighT blocks near the output. DeLighT matches the performance of transformers on standard low-, medium-, and high-resource corpora with 2-3 times fewer operations and parameters. We hope that our efforts towards significantly smaller and more data efficient deep learning models are impactful and valuable to the community.

---

> ### Comment · AnonReviewer5 · 2020-11-18
> **update**
>
> I would like to thank the authors for their extensive clarifications.
>
> It was important for me to see that
> (a) CUDA-less DeLighT trains faster than CUDA-less Transfomer
> (b) that the original Transformer paper already shows that using a smaller MLP or just 1 head does not do the job
>
> I am at this stage of the same opinion as Reviewer 1: there seems to be a sufficient evidence that DeLighT improves significantly upon Transformer, but the model's presentation in the paper is somewhat confusing. In particular, the emphasis should not be on depth/width/number of parameters, but on performance, computation and memory footprint. The updated paper however features good changes in the right direction. I am increasing the score.

---

### Author Response · Authors · 2020-11-11
**General response on computational efficiency**

We thank reviewers for their helpful comments. Most reviewers commented on the computational efficiency. We first address the comments related to computational efficiency here and then address reviewer specific comments.

For an apples-to-apples comparison, we implemented transformer unit without NVIDIA's dedicated CUDA kernel, and trained both transformer and DeLighT full-precision networks for 30K iterations on 16 NVIDIA V100 GPUs. **The transformer (67M parameters; 39.2 BLEU) and DeLighT (54M parameters; 40.5 BLEU) models took about 37 and 23 hours for training, respectively**. Significant engineering efforts from industry, such as dedicated CUDA kernels, have enabled very fast training of Transformers. DelighT doesn't yet benefit from dedicated CUDA kernels. For the same set-up, the transformer model with dedicated CUDA kernel took around 16 hours.

The main building block of DeLighT is group linear transformation, which can be parallelized. Similar to depth-wise convolutions in computer vision, group linear transformations (GLTs) can be parallelized. The solution that we used to implement GLT in this paper first splits the input into groups using reshaping and transpose functions (grouping function). Batch-wise matrix multiplication is then applied on each group to learn representations. Those groups are then combined with reshaping and transpose function (ungrouping function). To demonstrate that dedicated CUDA kernels can help improve the speed of DeLighT network, we implemented grouping and ungrouping functions with dedicated CUDA kernels. This decreased the training time from **23 hours to 19 hours**. Note that grouping, linear transformation, feature shuffling, and ungrouping can be implemented efficiently using a single CUDA kernel (e.g., ShuffleNetv2 uses a single CUDA kernel for shuffling and grouping). In future, we plan to implement a dedicated CUDA kernel for GLT using NVIDIA's general matrix multiplication library (CUTLASS). We believe that with a dedicated CUDA kernel for GLT, DeLighT would be much faster both during training and inference.

[r1] ShuffleNetv2: https://arxiv.org/abs/1807.11164

[r2] CUTLASS: https://github.com/NVIDIA/cutlass

---

### Author Response · Authors · 2020-11-18
**General response**

We thank the reviewers for their helpful comments. We would like to summarize the main changes we have made to the paper based on the reviewers feedback and discussion

1. We included Section 5 "Analysis and discussions on computation efficiency" that discusses the training time and memory consumption of Transformers with DeLighT and added Appendix E that contains the source code of GLT.

2. We updated Table 4 for a direct comparison with Transformer-XL.

3. We have included Figure 2 that illustrates DeLighT transformation with GLT, shuffling, and input mixer. We have also updated text in Section 3.

We have highlighted the changes in blue colored text in the new draft

---

### Decision · Program_Chairs · 2021-01-07
**Final Decision**

**Decision:**

Accept (Poster)

**Comment:**

This paper presents some innovations to transformers allowing some significant reductions in parameter count. While some reviewers were concerned that the proposed innovations seem incremental and may not stand the test of time, all reviewers recommended acceptance after engaging in a rich and interactive author discussion. Given the clear importance of making transformers more efficient I think this paper will be of interest to the community and is worthy of acceptance at ICLR.